# The Rustenburg Layered Suite formed as a stack of mush with transient magma chambers

Zhuosen Yao[1], James E. Mungall [1✉] & M. Christopher Jenkins [1]

The Rustenburg Layered Suite of the Bushveld Complex of South Africa is a vast layered accumulation of mafic and ultramafic rocks. It has long been regarded as a textbook result of fractional crystallization from a melt-dominated magma chamber. Here, we show that most units of the Rustenburg Layered Suite can be derived with thermodynamic models of crustal assimilation by komatiitic magma to form magmatic mushes without requiring the existence of a magma chamber. Ultramafic and mafic cumulate layers below the Upper and Upper Main Zone represent multiple crystal slurries produced by assimilation-batch crystallization in the upper and middle crust, whereas the chilled marginal rocks represent complementary supernatant liquids. Only the uppermost third formed via lower-crustal assimilation–fractional crystallization and evolved by fractional crystallization within a melt-rich pocket. Layered intrusions need not form in open magma chambers. Mineral deposits hitherto attributed to magma chamber processes might form in smaller intrusions of any geometric form, from mushy systems entirely lacking melt-dominated magma chambers.

[1] Department of Earth Sciences, Carleton University, 2115 Herzberg Laboratories, 1125 Colonel By Drive, Ottawa, ON K1S 5B6, Canada.
✉email: JamesMungall@cunet.carleton.ca

L ayered mafic intrusions represent portions of the plumbing systems of many large igneous provinces and are principal repositories of several critically important ore elements, including Cr, Ti, V, and the platinum-group elements (PGE)[1]. Layered mafic intrusions, such as the iconic Rustenburg Layered Suite (RLS) of South Africa, have historically been considered to represent the solidified remnants of vast liquid-dominated reservoirs of magma called magma chambers where crystallization-differentiation has occurred by fractional crystallization[2–4]. However, an emerging consensus in igneous petrology views magmatic plumbing systems as being dominated by interconnected bodies of mush (here we consider mush to be partially molten material containing anywhere from a few percent suspended solids with fluid-like rheology up to almost entirely solidified material with a yield strength) extending from the base to top of the lithosphere and only rarely containing more than a few volume percent of liquid at ephemeral and isolated locations[5–7]. It is still problematic whether mafic–ultramafic layered intrusions represent shallow, large-scale melt pockets or crystal-dominated mush zones within transcrustal plumbing systems.

The paradigm of magmatic evolution by fractional crystallization has dominated igneous petrology since Bowen's revolutionary advances a century ago[8], subject to recognition half a century later of the importance of crustal assimilation to result in the process of assimilation–fractional crystallization (AFC)[9]. The fundamental processes driving the evolution of magma composition in AFC are dissolution of host rock or xenoliths, accompanied by cooling and crystal growth, and the immediate removal of crystals from the possibility of continued reaction with the melt (Fig. 1a). Given that AFC is explicitly defined as a fractional process, it is inherent in all AFC models that the thermodynamically defined system at any given time is composed almost exclusively of melt, into which infinitesimal amounts of contaminant may be titrated, and out of which the consequent solids

derived by incremental crystallization must be removed. An AFC process may also be understood to occur via a reactive transport process where melt reacts with solids while migrating through a largely solid matrix and is driven to a new composition; this process may occur in deep crustal processing zones akin to the classical MASH (melting, assimilation, storage, homogenization) zone under magmatic arcs[9,10]. However when the concept of fractional crystallization is applied to layered intrusions, with or without a prior episode of assimilation, the conventional view is that magmatic evolution occurs within large, liquid-dominated melt reservoirs in the crust (i.e., magma chambers)[2,4]. This viewpoint has previously driven petrologists to search for the existence of the solidified remnants of such bodies in the rock record. A quintessential small example of closed-system fractional crystallization processes is the Skaergaard Intrusion of East Greenland[2,11]; however, the assumption that all large layered mafic intrusions must therefore also represent the solidified remnants of vast open chambers filled with melt is a reflexive model-driven extension of these ideas that has faced some recent challenges[12–15].

A simple conceptual alternative to AFC is that of assimilation in conjunction with the textbook process of batch, or equilibrium, crystallization (i.e., ABC; Fig. 1b). Although the conceptual differences between fractional and equilibrium crystallization may appear arbitrary and purely academic, they drive fundamentally different processes if they occur in large-scale magmatic systems. In the simplest expression of this concept, during ABC a magma becomes progressively more contaminated by the ongoing dissolution of wall rock or xenoliths; a hypothetical isenthalpic contamination drives the continuous crystallization of an increasing load of suspended solids which may remain broadly at equilibrium with the enclosing melt via intracrystalline chemical diffusion given sufficient time and high enough temperature[16]. As the proposed reason for fractional crystallization, the scenario of

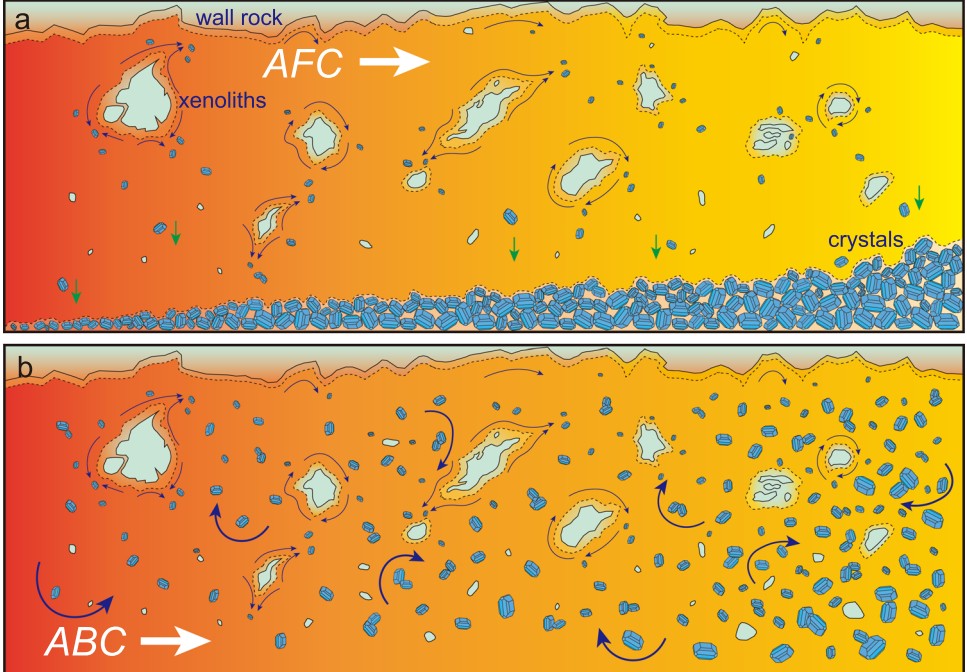

**Fig. 1 Schematic illustration of temporal evolution of magmas from left to right.** Bulk assimilation mostly occurs on the wallrock–magma boundaries via dissolution and/or from the crustal xenoliths induced by magmatic stoping. **a** Classical AFC model; heat for crustal assimilation is supplemented by concurrent fractional crystallization[8] while newly formed crystals are immediately sequestered from an almost entirely crystal-free liquid. **b** In the ABC model, during assimilation a steadily increasing amount of precipitated solids remains suspended by forced convection during magma flow and is continuously re-equilibrated with magma until it comes to rest and the solids are deposited all at once.

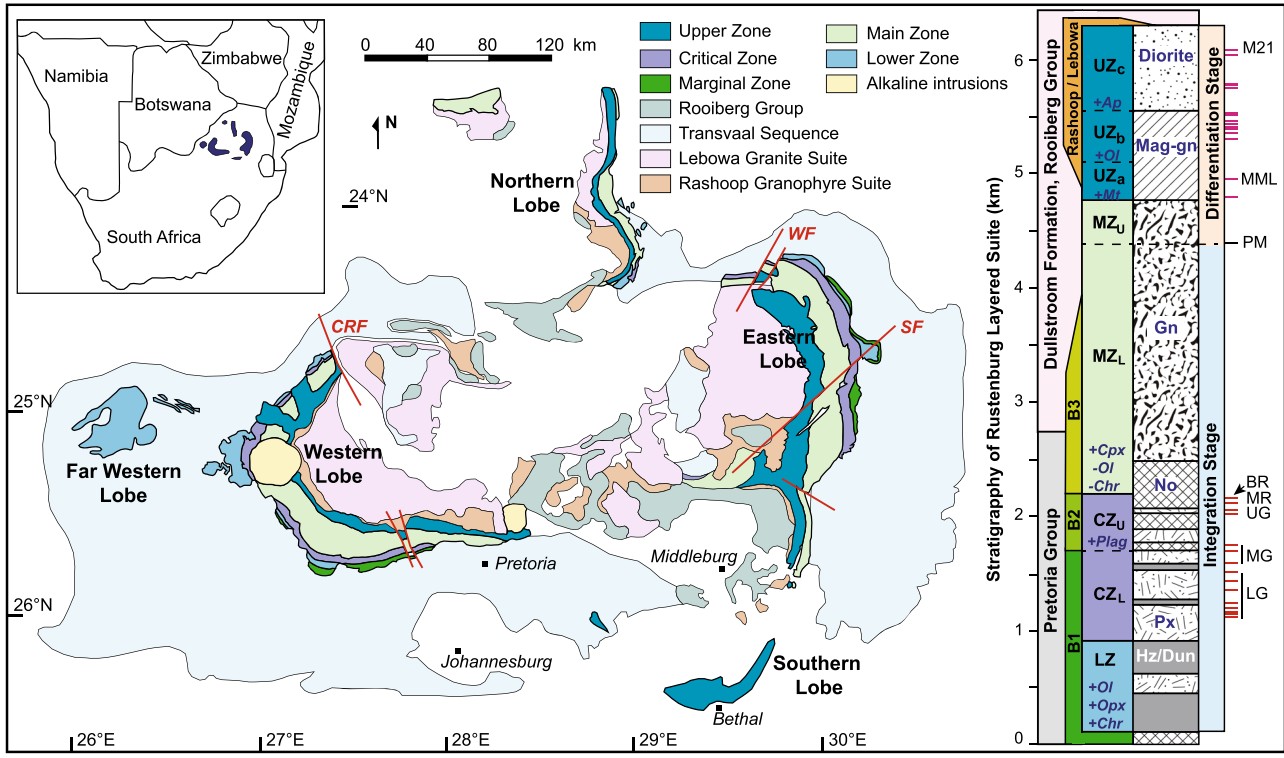

**Fig. 2 Geological map of the Bushveld Complex and stratigraphic column of the Rustenburg Layered Suite.** Adopted from multiple sources cited in the text[4,12,24,33,44]. Inset map shows the location of the Bushveld Complex within South Africa. Because the RLS is discordant with the underlying footwall rocks, its position within the host rock series varies from the base of the Pretoria Group in the north to the lower Rooiberg Group in the south. In addition, a small segment of the Main-Critical Zones occurs at higher stratigraphic levels near the Steelpoort fault, overlying the Dullstroom Formation of the Rooiberg Group. Marker horizons: LG lower group chromitites, MG middle group chromitites, UG upper group chromitites, MR Merensky Reef, BR Bastard Reef, PM Pyroxenite Marker, MML main magnetite layer, M21 magnetite layer 21. Abbreviation: LZ Lower Zone, CZ$_L$ Lower Critical Zone, CZ$_U$ Upper Critical Zone, MZ$_L$ Lower Main Zone, MZ$_U$ Upper Main Zone, Ol olivine, Opx orthopyroxene, Cpx clinopyroxene, Chr chromite, Plag plagioclase, Mt magnetite, Ap apatite, Hz harzburgite, Dun dunite, Px pyroxenite, No norite, Gn gabbronorite, Mag-gn magnetite gabbronorite, CRF Crocodile River Fault, SF Steelpoort Fault, WF Wonderkop Fault.

gravitational sinking of dense solids in static magmas[8] does not universally hold true in magmatic systems, e.g., crystallizing phases may not separate from an evolved magma with high viscosity caused by crystallization and SiO$_2$ enrichment. Additionally, heat exchange and crystallization in assimilation provide destabilizing buoyancy fluxes to drive forcefully disordered convection of mafic magma, where solids can be passively advected by vigorous convection instead of sinking[17]. In most turbulent komatiite flows, dense crystals can be carried by energetic eddies and remain in suspension without rapid separation[18]. Although equilibrium may not be attained at moderate temperatures, it is likely to occur rapidly in ultramafic magmas due to fast diffusions of elements in high-temperature, less-viscous liquids[19]. The success of a hypothetical dimensionless thermodynamic black box ABC process has been demonstrated numerous times, reproducing the observed compositions of ultramafic cumulate rocks and their supernatant magmatic liquids in intrusions both large and small[12,20–22].

A logical place to test the applicability of the ABC process to igneous petrogenesis at large scales is the RLS of South Africa (Fig. 2). Although much has been written about the genesis of the RLS, most published petrogenetic models represent qualitative interpretations motivated by the concept of the melt-dominated magma chamber[2–4]. Recent development of the concept of transcrustal, mush-dominated magmatic plumbing systems combined with geochronology[12] and thermal modeling of the RLS[14] invites the proposition that major parts of the RLS may have formed as mushes that were dominated by liquid only

during transient episodes of magma migration. Here we propose quantitative forward thermodynamic models that recreate the observed bulk rock, mineral, and isotopic compositions of all major constituents of the entire RLS via assimilation processes occurring throughout the crust. The spectrum of bulk cumulate macrolayer compositions observed in the RLS can be described as the first-order products of magma evolution during processes ranging from simple ABC in the upper crust for the ultramafic rocks of the Lower Zone and Critical Zone, through a two-stage ABC process in the mid-crust to generate the mafic rocks of the Upper Critical and Main Zones, to classical AFC in the lower crust to form the parental melt for the Upper and Upper Main zones, which then evolved by fractional crystallization in an essentially closed magma chamber affected by a small number of recharge events. We infer that the internal differentiation of macrolayers into subsidiary layers of different modal proportions, including monomineralic units like chromitites and anorthosites, resulted from second-order effects like crystal sorting during emplacement.

## Results

**Application to the RLS.** The Paleoproterozoic (~2.055 ± 0.001 Ga[12,23]) RLS is the world's largest layered mafic intrusive complex, containing ~600,000 km$^3$ of mafic–ultramafic cumulates and extensive reserves of PGE, chromium, and vanadium that dominate global resources of these elements[4,24]. The RLS intruded the 2.6–2.3 Ga sedimentary Pretoria Group and 2.061 Ga felsic lavas of the Rooiberg Group (~200,000–300,000 km$^3$) at upper-crustal

levels (~0.06–0.24 GPa)[24,25]. In conjunction with the overlying Rashoop Granophyre and Lebowa Granite Suites (~205,000 km³), they together constitute the Bushveld Complex, comprising an enormous bimodal continental large igneous province in the Kaapvaal Craton (Fig. 2). If there exists a larger magmatic plumbing system in the middle Kaapvaal crust beneath the RLS, its host rocks probably resemble the nearby Archean basement of amphibolite- to granulite–facies trondhjemitic–granodioritic–granitic gneisses, orthogneisses and metasedimentary rocks exposed in the Vredefort impact structure near Johannesburg[26] and in the Southern Marginal Zone of the Limpopo belt to the north of the RLS[27], whereas the information regarding the regional lower crust is highly limited in the literature.

The RLS is shaped like a dinner plate about 7–9 km thick and ~400 km in diameter (Fig. 2), with moderately inward-dipping marginal zones and flatter-lying central portions. Based on lithological and geochemical investigations, the RLS is traditionally subdivided into five major and laterally continuous stratigraphic zones (Fig. 2)[2,24,28]: (1) the fine-grained, noritic to peridotitic Marginal Zone (~100–750 m-thick), which flanks the other zones outside the main layered series and overlies a Basal Ultramafic Sequence (~750 m) encountered only in drill core beneath the other zones[29]; (2) discontinuous trough-like bodies of ultramafic Lower Zone (~800–1300 m) comprising harzburgite, orthopyroxenite, and minor dunite interlayers; (3) pyroxenitic Lower Critical Zone (~700–800 m) and noritic Upper Critical Zone (~500–1000 m), defined by the occurrence within both of them of prominent and laterally extensive chromitite and sulfide-bearing layers locally enriched in PGE; (4) gabbronoritic Main Zone (~3000–3400 m); and (5) uppermost ferrogabbroic-noritic and dioritic Upper Zone (~1700–2200 m) with abundant magnetite layers (Fig. 2). Moreover, the Main Zone is subdivided into the Upper (~300–700 m) and Lower Main Zone (~2300–3000 m) via a prominent, 3-m-thick orthopyroxenite which marks a significant reversal of initial Sr isotopic ratios (($^{87}Sr/^{86}Sr)_i$)[4,28]. The Upper Zone is also subdivided into three subzones by the first appearances of magnetite ($UZ_a$), olivine ($UZ_b$), and apatite ($UZ_c$) (Fig. 2)[30,31]. This broad zonal classification is oversimplified for its regional-scale utility, while the cumulate layers show numerous and complex mesoscale variations in their spatial distribution[4], e.g., intricate details of the lithological macrolayering in Lower Zone, Lower Critical Zone, and Upper Critical Zone (Fig. 2) which cannot be correlated regionally despite the apparent regional correlations implicit in the naming conventions used for the chromitite layers within them[32]. If the implicit regional correlation of the chromitites is correct, then its regional-scale uniformity is superimposed on a patchwork of locally variable layered cumulate rocks and the stratigraphic succession both below and above chromitite-bearing units is puzzlingly inconsistent from one section to another[32,33]. These local to regional-scale variations in the background against which the apparent regularity of the chromitites is imposed receive little explanation even when they are clearly documented. Additionally, the chromitites and some ultramafic layers have down-dip lithological and compositional variations, and also show transgressive contacts to their footwall rocks[24].

Within the prevailing conceptual framework that its layers were deposited in sequence from bottom to top, the overall stratigraphy of the RLS can be fitted into a two-stage pattern[4]. The formation of the lower portion from Lower Zone to Lower Main Zone has been referred to as the integration stage (Fig. 2), recording multiple influxes and extreme oscillations of dramatically different magmas at the scale of individual layers meters to tens of meters thick, as evidenced both by magmatic unconformities and sharp changes in lithology[24,34], and by the distribution of radiogenic isotopes, exemplified by ($^{87}Sr/^{86}Sr)_i$[4]. This is most clearly expressed in the Upper Critical Zone where variations in modal abundance within macrolayers such as the UG2 Unit result in juxtapositions of layers of anorthosite and norite with harzburgite, pyroxenite, and chromitite at scales of several meters[12]. However, zircon within these macrolayers has been shown in some cases not to have crystallized in a sequence younging upward through the stratigraphic column, an observation that has called into question the notion of a bottom-to-top aggrading crystal pile at the base of a long-lived magma chamber[12,34,35]. The formation of the upper portion of the RLS has been referred to as the differentiation stage (Fig. 2)[4], recording relatively uniform parentage as shown by marked uniformity of radiogenic isotope ratios and trace-element abundances against a backdrop generally considered to record a well-defined process of fractional crystallization in a large liquid-filled magma chamber with few magma recharge events[28,31,36,37], although some recent studies also tentatively attributed it to the emplacements of several batches of magmas with constant isotopic compositions[30,38]. The boundary between the integration stage and the differentiation stage is marked by the pyroxenite marker layer at the base of the Upper Main Zone (Fig. 2), and the resulting composite upper body postulated to have crystallized from a large, quiescent magma body can be termed the Upper and Upper Main Zone[31,37].

**Magma compositions.** Cumulate rocks in the complex can be subdivided into plagioclase-rich mafic units that have been postulated to be mafic crystallization products of tholeiitic magmas referred to as A-type magmas and ultramafic units that have been postulated to be products of high-MgO magmas referred to as U-type magmas[39]. The U-type magmas contained ~12–14 wt% MgO, corresponding to quench-textured norites exposed in the Marginal Zone surrounding the Lower Zone and Lower Critical Zone and referred to as the B1 marginal sills (Fig. 2)[25,40–42]. It has been presumed that the mafic sills of the Marginal Zone represent samples of the parental magmas that generated the RLS[3,25,40,41]. However, the most primitive olivine and orthopyroxene observed in the Lower Zone cannot be crystallized from the melts with the composition of the recognized B1 magma[24], and the mineral compositions and geochemical characteristics of the newfound Basal Ultramafic Sequence beneath the Marginal Zone require a komatiite as the true parental magma (>~19 wt% MgO)[29]. The Bushveld U-type magmas are compositionally similar to modern boninites formed by hydrous melting of metasomatized upper mantle[40], but Barnes[42] proposed a better analog in the siliceous high magnesium basalts derived from the crustal contamination of komatiites in Archean greenstone belts.

The A-type magmas, thought to have contained ~7–8 wt% MgO, are tentatively correlated with fine-grained gabbronorites of the Marginal Zone where it abuts the Upper Critical Zone and Lower Main Zone, respectively, termed the B2 and B3 tholeiitic magmas (Fig. 2)[39–41]. The origins of the A-type magmas have had less attention than that of the U-type, with most investigators apparently assuming that they are commonplace tholeiitic basaltic magmas[41] somehow derived from the upper mantle. Since the mantle does not directly produce tholeiites containing such low MgO contents, there must have been some processing of their parental magmas, though this process has not been clearly defined in the past[1]. B2 and B3 are also unconvincing parental magmas due to their partial cumulate characteristics and discrepant crystallization order compared to their interpreted cogenetic cumulates in the RLS[24,25]. The bulk composition of the lower portion of the RLS is too rich in compatible elements including Cr and the PGE to represent the composition of a liquid—it is necessarily regarded as being composed of cumulates

deposited from larger volumes of through-going magma that are not presently exposed within the RLS[43].

The bulk composition of the parental magma that was injected to form the Upper and Upper Main Zone has been modeled by adding ~15–25% of a hypothetical missing segregated component into a weighted average Upper and Upper Main Zone bulk composition, to form a basaltic andesite with ~4–6 wt% MgO[28,37], but a modeled fractional crystallization sequence from this magma still does not closely resemble the natural occurring cumulates[37].

Recently discovered spinifex olivine margins chilled at the base of the Lower Zone[44] or from the Basal Ultramafic Sequence[29] strongly argue for a komatiitic parent magma that, at least locally, was chilled against the quartzitic floor. From this perspective, the remarkable similarities of mantle-normalized trace-element patterns have led to suggestions that the B1 and B2-3 magmas were derived from komatiite via >~40% contamination of upper and lower crust, respectively[12,25,44]. Sr- and Nd isotopic data of cumulates were used to support the proposition that primitive melt assimilated ~15–30% partial melt of upper crust to produce the Lower and Lower Critical Zones, whereas ~40–50% contamination with the depleted restite in a staging chamber beneath RLS is required for the Upper Critical and Main Zones[45]. Cr enrichment and cyclic compositional reversals in the Lower Zone have been attributed to episodic influxes of crystal+liquid slurries derived from komatiite contaminated by 20% crust at 0.45–1.0 GPa[43]. Consideration of the Cr budget during chromitite formation indicates that the parental liquids must have been komatiitic[12,22,43]. Contamination in deep-seated chambers before final crystal-slurry-type emplacement into the RLS was also proposed on the basis of stable and radiogenic isotope systems[27,46].

In contrast to the various suggestions of crustal contamination, radiogenic $^{187}Os/^{188}Os$ of sulfide[47] and unradiogenic $\varepsilon_{Hf}$ of zircon from the RLS[48] have been used to suggest that the parental magmas instead inherited their lithophile element compositions from ancient eclogite-bearing subcontinental lithospheric mantle. However, newer data have demonstrated a limited range of Hf isotopic composition in both the RLS and its plausible local crustal contaminants[49]. The Zr–Hf budget and associated unradiogenic $\varepsilon_{Hf}$ in the RLS require the addition of crustal components[49].

A role for refractory subcontinental lithosphere was also proposed as a possible explanation for the exceptionally high Pt/Pd of Bushveld U-type magmas and mineral deposits[50], but it must be noted that a large degree of melt production is highly unlikely from relatively cool and previously melt-depleted subcontinental lithosphere[51]. Partial melting of an ancient subduction-affected, eclogitic component in subcontinental lithosphere can produce a siliceous basalt similar to the B1 magma, but its consequent MgO content cannot exceed 15.5 wt% even after ~79% melting in high-pressure experiments[52]. Since the alleged subcontinental lithosphere Os isotopic signature could equally well be derived from crustal contaminants[53], and melt–rock reaction by asthenospheric melts while they pass through refractory subcontinental lithosphere might boost Pt/Pd, in the balance we favor the idea that the massive and very short-lived injection of magma that formed the RLS resulted from rapid melting of an asthenospheric mantle plume for the Bushveld large igneous province.

Considering the apparent necessity of a komatiitic parental magma for the RLS, we suggest that rather than representing samples of the magmas parental to the RLS, the sills preserved in the Marginal Zone may instead be samples of magma that had already passed through the complex, depositing layered cumulate rocks within the RLS before their eventual expulsion into the surrounding Pretoria Supergroup[12]. Regarding the marginal zone magmas as the complements to the cumulate rocks rather than as their parents alleviates some of the more serious mass balance concerns. Given the possibility of a transcrustal magmatic plumbing system beneath the Bushveld large igneous province, assimilation of crust by komatiitic magmas may have occurred at upper-, middle-, and/or lower-crustal levels, which is also proposed by geochemical investigations mentioned above[12,25,27,43–46].

**Thermodynamic modeling.** To test the applicability of ABC and AFC to the petrogenesis of the RLS, we have modeled the processes using alphaMELTS thermodynamic software[54], supplemented by models of isotopic mass balance constrained by the alphaMELTS results. The working hypothesis was that it might be possible to produce representatives of each cumulate rock type preserved as individual macrolayers in the RLS by transcrustal assimilation processes. We chose to model average compositions for each of several major mafic–ultramafic lithologies and the B1-3 marginal sills (Fig. 3) on the assumption that, once contaminated magmas had formed with compositions close to average compositions of the major units of the RLS, grain sorting on the macrolayer or hand specimen scale led to internal differentiation of the larger macrolayers into sublayers having widely varied modal proportions of the incoming minerals, accounting for the existence of some monomineralic rocks and for much of the observed scatter about the mean values[22].

We have considered two distinct scenarios to address the possible origins of the integration stage cumulates beneath the base of the Upper and Upper Main Zone and a third for the Upper and Upper Main Zone, illustrated in Fig. 3. In Scenario 1, following the one-stage ABC approach we have already successfully applied to several ultramafic suites worldwide[12,20–22], komatiite is combined with a upper-crustal assimilant in an isenthalpic process, creating a relatively cooler equilibrated mixture of liquid and crystals which then undergoes some degree of cooling while remaining internally at equilibrium. Given the extremely low viscosity (~0.05–0.2 Pa s), ascent rate as great as m/s and high liquidus temperature (>1550 °C) of komatiite[18], its emplacement into cooler host rocks (~200 to 300 °C) approximates to forced turbulent convection during the early assimilation process, where solids are passively advected by chaotic flow and remain in suspension (Fig. 1b)[17,18]. This first stage represents a single batch process of assimilation and cooling during shallow transport. After the flow is emplaced into a sill-like body at the level of the RLS, the vigorous momentum of turbulent flow is dissipated by assimilation and emplacement, and the crystal–melt mixture is eventually separated by gravity into a cumulate comprising mostly solids and some trapped liquid, and a supernatant magma comprising mostly liquid and some entrained solids. During the resulting dumping of most of the entrained crystal load to form a macrolayer, internal layering forms in a manner analogous to the stratifications of bed-load sediment in water in upper plane bed flow regimes. These successive sill-like magma pulses can be vertically stacked in any order to build up a thick layered pluton consistent with field observations, geophysical data, and numerical models[55]. Hence, the model cumulate in this upper-crustal assimilation stage is compared with ultramafic cumulates of the RLS, and the supernatant magma leaving the system is compared with B1 marginal sill compositions.

In Scenario 2, representing two successive batch steps, it is assumed that a first ABC process occurs in a mid-crustal reservoir, after which the supernatant liquid rises and undergoes a second batch crystallization as it cools and is emplaced at the level

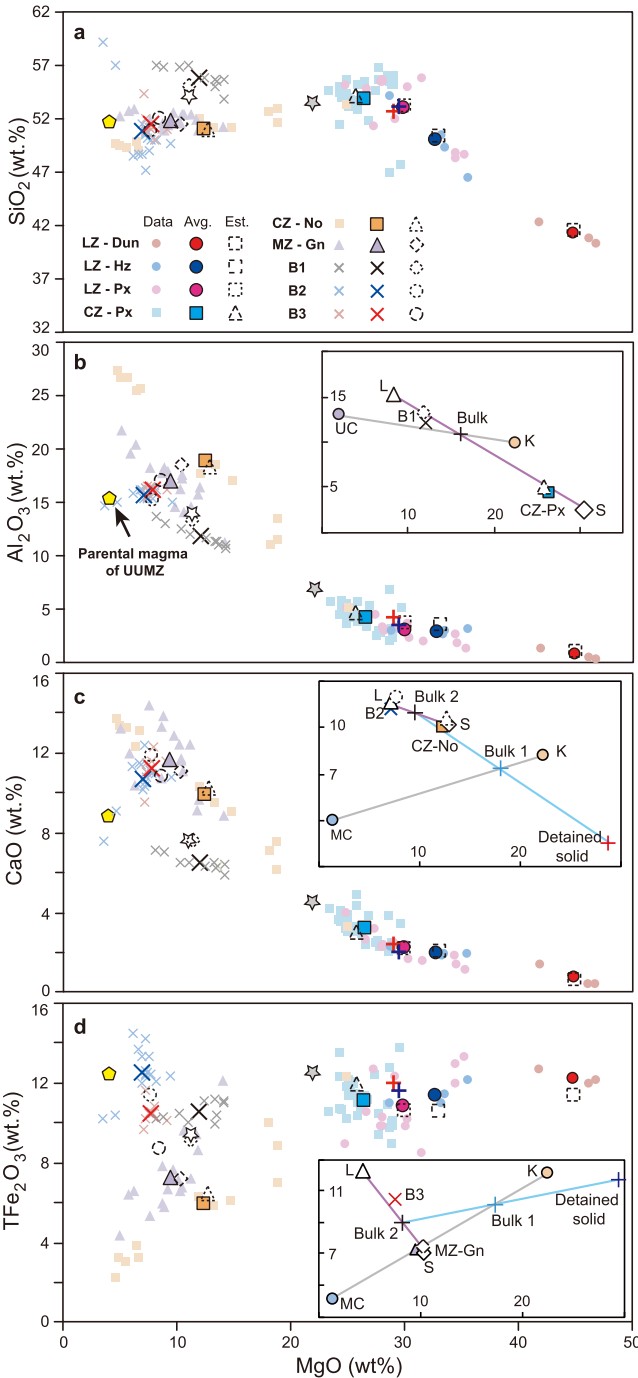

**Fig. 3 Comparisons of whole-rock geochemical data with the model results.** Bivariate diagrams to show the variations of **a** SiO₂, **b** Al₂O₃, **c** CaO, and **d** total Fe₂O₃ versus MgO for the whole-rock compositions from the lithogeochemical section of the RLS[24]. The legends for different lithological units are shown in **a**. Compositions of komatiite, upper crust, and middle crust are shown as K, UC, and MC, respectively. Gray and purple lines represent the komatiite-crust assimilation trends and the solid–liquid reallocation relationship, respectively, in the B1-Px (Critical Zone) lineage (**b**). Similar relationships exist for each of the ABC processes to generate other Lower Zone and Critical Zone ultramafic macrolayers (Supplementary Fig. 1). The two-stage ABC processes creating the B2-No (Critical Zone) lineage and B3-Gn (Main Zone) lineage are illustrated in the insets of panels **c** and **d**. A first ABC process generates liquid and solid, and the compositions of solids retained at depth for the B2-No (Critical Zone) and B3-Gn (Main Zone) lineages are marked as vertical red and blue crosses, respectively. Removal of first-stage solids as described in the text generates a new bulk composition (Bulk-2) that is emplaced in the RLS and crystallized to form the illustrated solid and liquid compositions. Inset diagrams zoom in on the specific area where crystal sorting between model liquid (L) and solid (S) would generate the observed cumulates and corresponding phenocryst-bearing marginal rocks. The white and gray pentagrams represent the modeled B1 and UG2 cumulate, respectively, from the one-stage ABC approach of Mungall et al.[12]. The proposed parental magma for the Upper and Upper Main Zone (UUMZ) is exhibited as yellow pentagons, and also shown in the Supplementary Table 3. Hz harzburgite, Dun dunite, Px pyroxenite, No norite, Gn gabbronorite, Ave. average compositions of the corresponding lithologies based on the representative section of RLS from Maier et al.[24], Est. estimated compositions derived from the thermodynamic models. All data collected from multiple sources cited in the text[24,25,40,41] and provided as a Source Data file.

approximate to AFC as classically understood, presumably occurring during melt percolation through a complex lower-crustal magma reservoir that may have comprised multiple interconnected sill- and dike-like bodies largely composed of mush[5–7]. Liquid that has been processed through this AFC mush zone is extracted and emplaced into a sill-like magma chamber where it subsequently evolves by fractional crystallization, subject to some subsequent magma replenishment events during the formation of the Upper and Upper Main Zone[31].

The parameters used in the models are provided in Supplementary Tables 1–4. Compositions of endmember magmas, contaminants, solids, liquids, cumulates, and ejected magmas are all shown in Figs. 3–6. The parental mantle-derived melt is an Al-undepleted komatiite[12]. Major and trace element and Sr, Nd, and O isotopic compositions of the magmas and contaminants were estimated by comparison with upper-crustal and mid-crustal rocks exposed in the Pretoria Supergroup, Vredefort impact structure, and Limpopo Belt as documented in detail in Supplementary Table 4 and Supplementary Figs. 3 and 4.

Ultramafic cumulates of the Lower Zone, Lower Critical Zone, and B1 marginal sill compositions were modeled under Scenario 1, assuming an upper-crustal assimilant at 0.2 GPa[24,25], following a previous Scenario 1 model for the UG2 pyroxenite of the Upper Critical Zone and complementary B1 magma[12]. After assimilation of 17.4% upper crust, Lower Zone dunite could form as an adcumulate comprising 4.5% trapped liquid; Lower Zone harzburgites require 22.5% assimilation and are modeled as mesocumulates comprising 17.8% trapped liquid, whereas Lower Zone and Lower Critical Zone pyroxenites could have formed after 27–34% assimilation of upper crust to leave a cumulate containing ~15% trapped liquid (Supplementary Table 1 and

of the RLS to form a mushy macrolayer (Supplementary Fig. 2). During emplacement, this new batch of crystals and melt then separates into cumulates comprising mostly solids and some trapped liquid to represent part of the RLS, and a supernatant magma comprising mostly liquid carrying some entrained crystals that can be compared with the marginal sills. Compositions of solids, liquids, cumulates, and marginal sills are shown in Fig. 3c, d. This second scenario is therefore a sequence of two batch equilibrium processes which allows for the separation of hidden cumulates from the bulk mixture prior to magma ascent and deposition of the cumulate layer in the RLS—it might be considered as the first step along a continuum of possible processes toward AFC.

In Scenario 3, the assimilation and batch removal of crystals occur in a large number of small steps (e.g., 20–50 steps) that

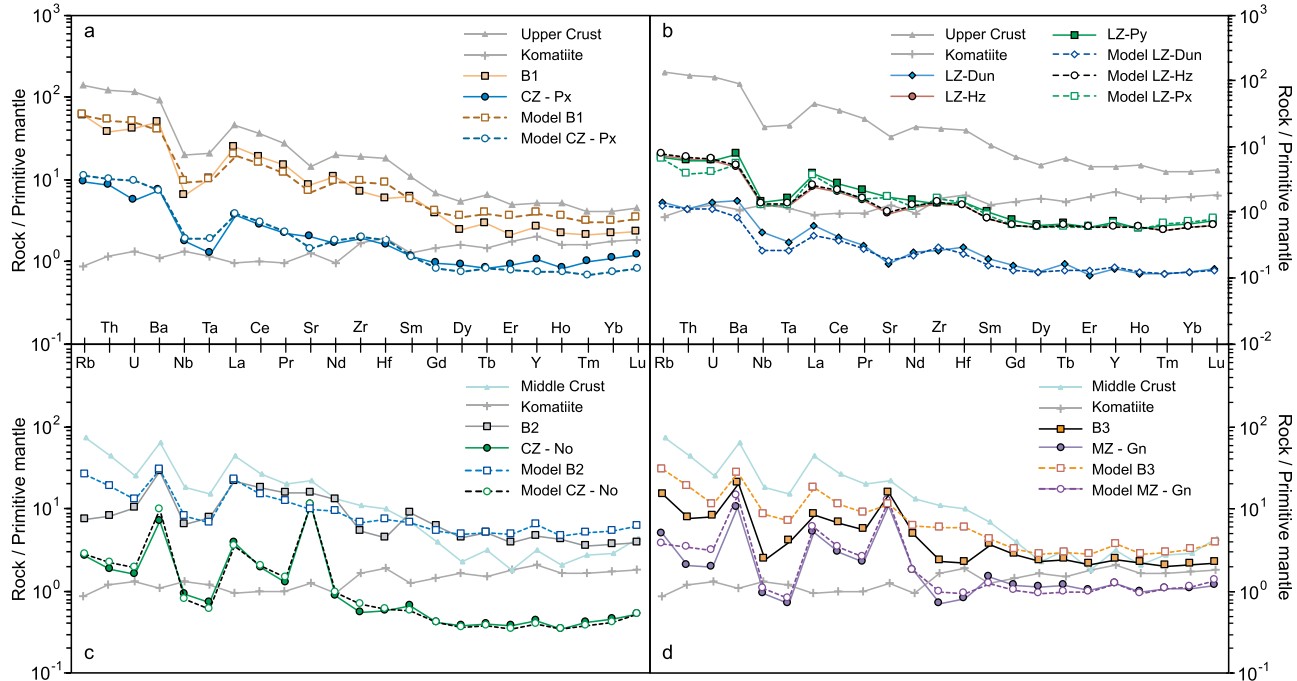

**Fig. 4 Primitive mantle-normalized trace-element concentrations for natural and model rocks from the RLS.** Solid symbols joined by solid lines represent average compositions of rocks from the RLS, while open symbols joined by dashed lines represent alphaMELTS models. The ABC model accounts for the trace-element patterns of **a** B1-Px (Critical Zone) lineage and **b** other Lower Zone and Critical Zone ultramafic macrolayers. Additionally, the trace-element compositions of the **c** B2-No (Critical Zone) lineage and **d** B3-Gn (Main Zone) lineage coincide with the modeling results of two-stage ABC processes. Compositions of upper crust, middle crust, and komatiite are shown in Supplementary Table 3. Source data are provided as a Source Data file.

Supplementary Fig. 1). The B1 magma is modeled as a mixture of 22% solids equivalent to Lower Critical Zone pyroxenite with 78% liquid. The trace-element compositions of these cumulates and B1 marginal sills coincide with the modeled results (Fig. 4a, b). Because the B1 marginal sills that envelope the Lower Zone and Lower Critical Zone of the RLS range in thickness from 100 to 400 m and can further penetrate ~100 km into the floor rocks[40,41], their total volume may be regarded as supernatant magmas complementary to emplacement of all of the Lower Zone and Critical Zone pyroxenites. If the AFC model is adopted under the same settings, the peritectic reaction between olivine, orthopyroxene, and liquid forbids the formation of the commonly observed coexistences of olivine and orthopyroxene in cumulates (i.e., granular harzburgites), while also failing to match observed whole-rock compositions.

Mafic rocks of the noritic portions of the Upper Critical Zone and gabbronoritic Main Zone are modeled under Scenario 2, with the same komatiite parent melt but a mid-crustal assimilant at 0.45 GPa that corresponds to the mean depth of middle continental crusts. The corresponding temperature of the contaminant was estimated as 390 °C via the geothermal model for continental lithosphere and a higher heat flow for the Paleoproterozoic RLS (~70 mW m$^{-2}$) than its current value (51 ± 6 mW m$^{-2}$)[56]. The noritic Upper Critical Zone is xenolith rich, and has widely been regarded as an independent sill-like intrusion of progressive mixtures between B1 and B2/B3 magmas[24,25]. The similar crystallization sequence of Lower Main Zone also requires mixed parental magmas that intrude as crystal slurries from a deeper, staging reservoir after crustal assimilation[24]. We envision that their primitive parental komatiites experienced ABC assimilation (~21% for upper Critical Zone and ~24% for Lower Main Zone) and cooled to ~1240–1250 °C in the middle crust to obtain the Bulk-1 compositions (Fig. 3c, d and Supplementary Fig. 2). Retention of ~90–97% of the solids at the

site of assimilation left hidden ultramafic cumulates (e.g., the detained solid in Fig. 3c) in the middle crust with compositions very similar to the Lower Zone pyroxenites (Fig. 3). The remaining solids and liquid (Bulk-2 in Fig. 3c, d and Supplementary Fig. 2) were cooled by conduction during ascent and emplacement and then separated at the level of the RLS into mafic cumulates and ejected supernatant magmas very similar to the B2 and B3 marginal sills. After 40% crystallization at 1181 °C adcumulate norite in the Upper Critical Zone contains only 5% trapped liquid; its ejected liquid complement with only 5% solids resembles the B2 magma apart from the depletion of Rb and Th in the B2 composition (Figs. 3 and 4c). After 63.9% crystallization at 1130 °C the modeled Lower Main Zone magma settles to form a mesocumulate containing ~3% trapped liquid and is flanked by marginal B3 magma that is ejected at a relatively low crystallization degree (~34.2%) and contains ~42% solids (Figs. 3 and 4d). Relatively slow cooling and accumulation of crystals may account for coarser grain sizes and low trace-element abundances of the B3 (Fig. 4).

Scenario 3 is applied to the genesis of the Upper and Upper Main Zone. The occurrence of numerous titanomagnetite layers within Upper and Upper Main Zone (Fig. 2) indicate that the incoming parental magma was iron rich[28,37], and hence mafic lower crust is favored as the assimilant[57]. We suggest that slow rates of introduction of primitive magma into hot lower crust (assumed as 770 °C at 1 GPa due to the heat flow of ~70 mW m$^{-2}$) after passage of the vast volumes of magma that produced the Main Zone, combined with muted temperature gradients, might have permitted efficient crystal separation during ongoing crustal assimilation[17] (Fig. 1a) or alternatively a reactive transport process in a mush-dominated lower-crustal reaction zone, in contrast to the vigorous forced convection that favored crystal entrainment during formation of U-type magmas at shallow depth. We modeled the genesis of the parental magma of the

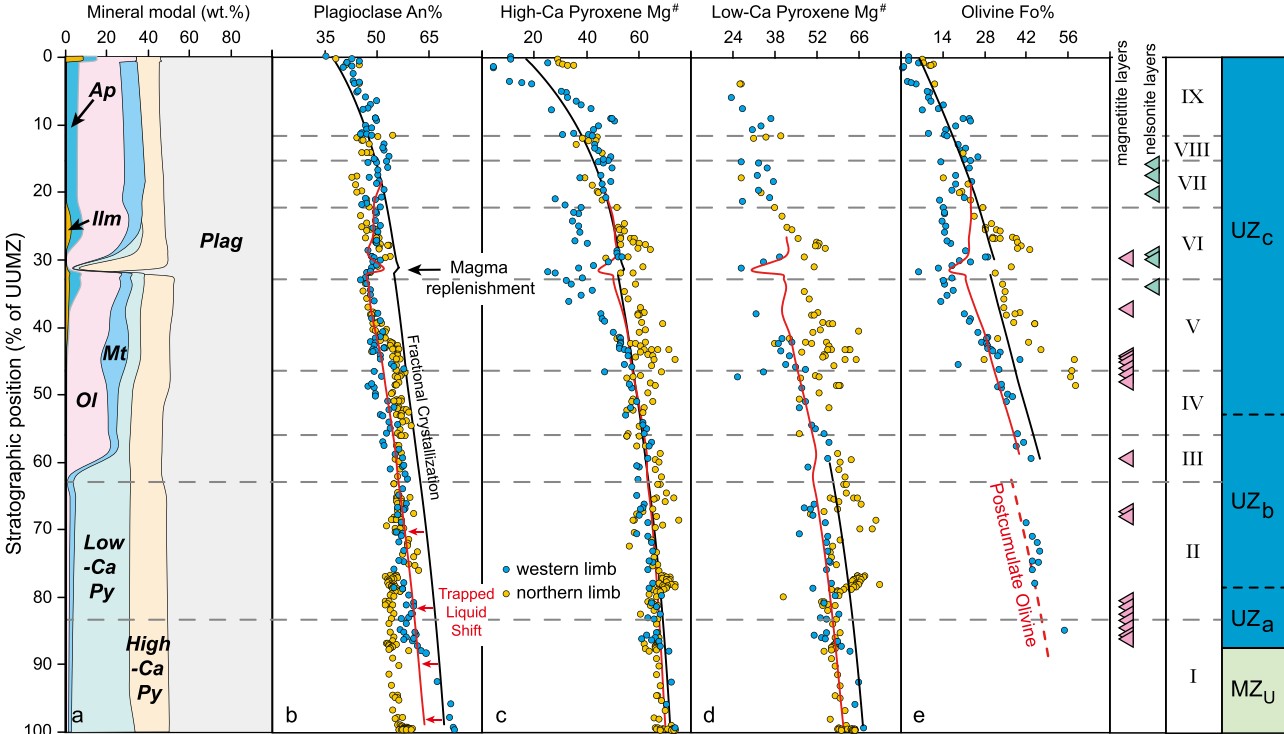

**Fig. 5 Modal abundances and compositional variations of minerals throughout the Upper and Upper Main Zone. a** The predicted mineral modes from AlphaMELTS modeling for the proposed parental magma of Upper and Upper Main Zone in Fig. 3. Comparisons between the measured composition variations (blue circles from the Bierkraal at the Western Limb[28,31]; yellow circles from the Bellevue at the Northern Limb[30]) and model results of **b** plagioclase (An%, 100Ca/(Ca + Na)), **c** high-Ca pyroxene (Mg#, 100 Mg/(Mg + Fe)), **d** low-Ca pyroxene (Mg#) and **e** olivine (Fo%, 100 Mg/(Mg + Fe)) with stratigraphic position. Cycles I–VI are identified by marked reversal in An% number of plagioclase from the Bierkraal drill cores, western limb[28]. A further three cycles (VII–IX) are defined by the disappearance of apatite without apparent reversal in An%, but have still been explained in the same way as cycles I–VI[28]. The Upper Zone contains ~30 magnetitite and nelsonite (magnetite–ilmenite–apatite cumulate) layers that hosts world-class V, Ti, and P sources. Compositional variations of major minerals crystallized from the incoming parental magma are shown as the black lines, and meanwhile the trapped liquid shifts (red lines) represent compositional modifications of minerals induced by the reduction of trapped interstitial melt fraction from 25 to 5% in the post-cumulate stage. Olivine in cycle II is crystallized from the intercumulus melt in this stage, corresponding to its low modal proportion (~1–2%) and prismatic shape[31]. Replenishment by ~1.2% initial parental magma at the boundary of cycles V and VI drives apparent reversals of mineral modes and compositions, which coincide with the observed data (reversals in olivine Fo%, pyroxene Mg#, and plagioclase An%)[28,31]. Plag plagioclase, Px pyroxenite, Ol olivine, Mt magnetite, Ilm ilmenite, Ap apatite. Source data are provided as a Source Data file.

Upper and Upper Main Zone (Fig. 3) by a process of 43.5% AFC contamination in the lower crust plus a further 24% fractional crystallization during slow upward ascent. After emplacement of this magma in the upper crust, the observed paragenetic sequence and mineral modes of the cumulate rocks can be reproduced via a closed-system fractional crystallization model until ~21% melt remains (Fig. 5a). Despite its overall success, the simple fractional crystallization model cannot account for the enigmatic magnetitite layers of the Upper Zone, which may require the operation of exceptional and poorly constrained processes such as double diffusive convection or liquid immiscibility[28,58]. It is debatable whether the final residual liquid was then erupted to form the upper portions of the Rooiberg felsites[4,37], but resolution of this controversy is not material to the success of our models because they focus on magma sources, not on their final residues. Compositional variations of major minerals throughout the Upper and Upper Main Zone are fitted if the trapped-liquid-driven compositional shift is included (Fig. 5), but also exhibit a series of minor reversals driven by several batches of magma replenishment, which has also been suggested to result in the formation of magnetitite layers[31,38]. The largest reversals of plagioclase An% (An$_{45}$ to An$_{51}$, Fig. 5b), clinopyroxene Mg# (~26–53, Fig. 5c), orthopyroxene Mg# (~27–42, Fig. 5d), and olivine Fo% (Fo$_6$ to Fo$_{29}$, Fig. 5e) across the boundary between

cycles V and VI, for instance, can be modeled by a small-scale (~1.2%) magma replenishment (Fig. 5).

We have modeled isotopic compositions of the cumulate rocks by tracking isotopic mass balances in the various mixtures of primary magma and assimilants according to the alphaMELTS models. The isotopic compositions observed in the RLS are compared with the results of our model of the transcrustal assimilation processes in Fig. 6. The measured inverse correlation between ($^{87}$Sr/$^{86}$Sr)$_i$ and $\varepsilon_{Nd}$ values of RLS (Fig. 6a) is matched well by all models except for the B3 magma, which is represented by very few samples[25]. Restricted ranges of ($^{87}$Sr/$^{86}$Sr)$_i$ in B1, B2, and B3 marginal sills have been widely used to support assertions that these were samples of the U-type and A-type parental liquids of the RLS, but these observations are equally consistent with our proposition that the sills represent the liquid residue from deposition of the corresponding cumulates (Fig. 6a). The newfound in situ Sr isotope disequilibrium between coexisting minerals and within plagioclase crystals from the RLS can be attributed to the mixing and sorting of variably contaminated crystal grains and/or local percolations of multiple, isotopically distinct melts in the post-cumulate stage[59], which has recently been demonstrated in the Rum layered intrusion[15]. High $\delta^{18}$O (average 7.1‰) in the RLS without apparent systematic changes is consistent with isotopic compositions of the proposed crustal

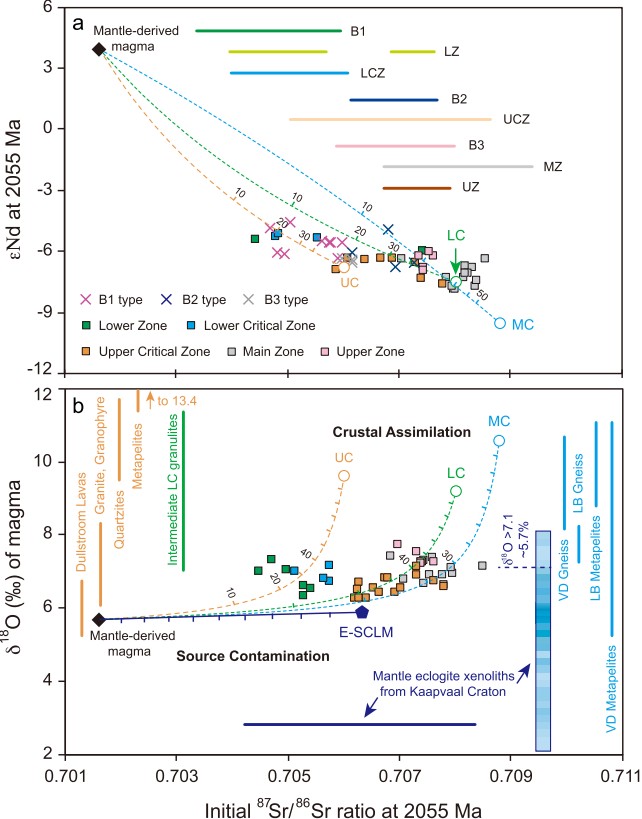

**Fig. 6 Isotope correlation diagrams comparing RLS rocks to assimilation models.** Curves represent modeled isotopic mixing between primary, mantle-derived magma and different end-members from the Kaapvaal Craton: dashed orange line, upper crust; dashed blue line, middle crust; dashed green line, lower crust; purple line, eclogite-bearing subcontinental lithospheric mantle (E-SCLM). **a** $\varepsilon_{Nd}$-($^{87}Sr/^{86}Sr)_i$. Horizontal bars in the top-right region exhibit the larger $(^{87}Sr/^{86}Sr)_i$ ranges that also include the Sr isotope compositions of samples and minerals without the coupled Nd isotopic data in marginal rocks and cumulate zones of RLS. **b** $\delta^{18}O$-($^{87}Sr/^{86}Sr)_i$. $\delta^{18}O$ averages with standard deviations of potential assimilants in upper, middle, and lower crusts are shown schematically by vertical orange, blue, and green bars, respectively; averages for various contaminants are shown by circles and pentagram. $(^{87}Sr/^{86}Sr)_i$ average plus its standard deviation for eclogite-bearing lithospheric mantle is exhibited by purple horizontal bar, and $\delta^{18}O$ distribution of mantle eclogite xenoliths from the Kaapvaal Carton is shown by vertical blue color bar with deep tone corresponding to a higher frequency. Garnets with $\delta^{18}O > 7.1$‰ represent only ~5.7% of the total of 157 available samples from Kaapvaal Craton[60]. All data are collected from multiple sources cited in the Supplementary References. LB Limpopo Belt, VD Vredefort Dome. Source data are provided as a Source Data file.

assimilants (Fig. 6b), but not with the composition of the eclogite-bearing lithospheric mantle of Kaapvaal Craton (mode $\delta^{18}O = $ ~5.9‰) which contains some of the most $^{18}O$-depleted (<4.5‰) garnets in the global database[60].

## Discussion

Our results show that the bulk of the RLS below Upper Main Zone, about 2/3 of its total thickness, appears to have been generated by either one-stage or at most, two-stage episodes of batch assimilation and crystallization and emplacement in their current locations as crystal mushes (Fig. 3). The melts left over from these processes can be represented by the marginal sills.

The application of the ABC concept to magmatic systems in lieu of AFC requires a fundamentally different perspective on the physical form of the magmatic systems in space and time and relaxes some constraints that would be imposed by the idea of fractional crystallization. For crystals to be able to re-equilibrate continuously with the melt during ABC it must be very hot and less viscous, tending to favor the process in ultramafic magmas but less so in mafic magmas. Furthermore, they must remain suspended and the melt must be well-mixed; both conditions require that the system is undergoing vigorous convection[17] and/ or turbulent flow[18], where the fluid dynamics is dominated by inertial forces and violent swirls/ eddies in both vertical dykes and sill-like bodies during magma transport and emplacement. Free, smooth convection or laminar flow cannot accomplish this, especially in offering enough vertical component of flow velocity to offset the settling of dense grains. Free magma convection in a hot sill emplaced between cooler host rocks is sluggish and entirely driven by the descent of cool crystal-laden drips to a stagnant base[61,62]. Once they reach the cool lower boundary, crystals cannot be re-entrained in the convective flow. Except in the exceptional case that a mafic or ultramafic magma reservoir is being heated from below, the requirement of vigorous stirring instead demands that the process is occurring as forced convection in a dynamic flowing magmatic setting like a network of dikes and sills[5–7]. Confinement to a dynamic conduit setting therefore also implies that ABC occurs quickly during transit of magma through the lithosphere rather than during quiescent evolution of a large melt-dominated magma chamber. It is implicit in an ABC model that as soon as the magma comes to rest, dense crystals will separate from the melt, arresting the process and forming masses of cumulates at any point where magma velocity slows due to the energy expenditure via assimilation and viscous dissipation related to emplacement.

The notion of complete internal chemical and isotopic equilibrium is a convenient starting point to consider ABC processes; however, the ABC concept is also able to accommodate observations of isotopic disequilibrium between crystals in a macrolayer[46,63]. If the crystals were amassed from sites along a dynamic transcrustal conduit where the isotopic properties of assimilants vary widely but the major element compositions and mineral modes were driven by the same fundamental reactions, then a variety of isotopic compositions is to be expected unless the system is given a long time to homogenize before deposition of the slurry. The alternative notion of slow fractional crystallization from a well-mixed and homogeneous reservoir of liquid in a long-lived magma chamber seems to demand a high degree of isotopic homogeneity within a given layer.

AFC and ABC therefore offer extremely different views on the mechanism of delivery of crystals to layered intrusions and consequently on the mode of formation of the intrusions themselves. In classic fractional crystallization models the crystals form slowly in small numbers in cool zones near the margins, either settling[2,8] or remaining in situ[62], to form layers, whereas in ABC the crystals form rapidly during transit through the lithosphere and are dumped in intrusions as masses that may subsequently undergo some crystal sorting into layers[12,22,24,64,65]. There can be no doubt that both mechanisms operate, exemplified by the record of fractional crystallization in, e.g., the small box-like Skaergaard Intrusion[2,62], and that of batch emplacement of crystal-rich loads in, e.g., olivine-rich Hawaiian picrite lavas[66]. Assembly of a large volume of mush through multiple emplacements of magmas generated by the ABC process is a viable alternative mechanism for the creation of a thick accumulation of mafic or ultramafic crystal mush that will later be recognized as a layered intrusion[55]. This mechanism crucially does not require the layers to have been emplaced in a younging-upward series at

the bottom of a classic magma chamber and accommodates recent geochronological[12,13] and field[34,35,67] evidence for out-of-sequence layer formation in major layered mafic intrusions and the brittle emplacement of residual melts extracted from the Upper Critical Zone into the overlying Main Zone[68]. It is also consistent with the contradictory observations of regionally correlated chromitite-bearing macrolayers that occur within host silicate cumulate sequences that cannot be correlated over the same regions[33,34], if the chromitite-bearing macrolayers were intruded as sheets within older and regionally variable cumulates.

Wholesale wallrock assimilation and thorough internal equilibration is difficult or impossible for multiply-saturated basaltic magmas, in which large degrees of solidification are experienced over small ranges in liquidus temperature. In contrast, hot and primitive MgO-rich magmas like komatiites are able to assimilate relatively fusible crustal rocks, including granitoids, basalts, and common sedimentary rocks, in large proportions, because their liquidus surfaces are very steep, aided by the latent heat of fusion liberated by the simultaneous crystallization of large volumes of the mafic minerals olivine and pyroxene[12,20,69], especially if the crustal rocks are already hot. Indeed, if crustal rocks are hot enough, they act as a solvent and can be added without limit to a komatiite without ever causing the system to solidify fully unless it is cooled. This is true regardless whether the process is one of AFC or ABC. A typical komatiite melt with as little as 18 wt% MgO can assimilate masses of warm crustal rock exceeding 50% of its original mass, generating a mass of cumulus olivine and pyroxene approximately equal in mass to the original mass of assimilant[12,20,69]. The resulting contaminated magma will therefore comprise approximately one-third ultramafic solids and two thirds low-MgO basaltic liquid. This solid fraction is well within the range of mobile crystal suspensions that can travel through the crust with essentially Newtonian rheology and density lower than most crustal rocks[70]. The assimilation process occurs so easily that uncontaminated komatiites are rare, and it works to prevent the existence of superheated magmas, which cannot fail to react with and dissolve their containers of host rock. Meanwhile, the extremely high temperatures and low viscosities of komatiites easily drive fast ascent and turbulent flow during emplacement, in which the dense crystals remain in suspension and at equilibrium with the host magma[18]—approximating to the conceptual ABC model.

It is especially noteworthy that modal proportions of cumulus minerals in ultramafic cumulates such as olivine–chromite or olivine–orthopyroxene mixtures in layered mafic intrusions generally do not conform to the instantaneous cotectic modal proportions expected during fractional crystallization, positively requiring that many such cumulates were deposited and mechanically sorted into layers from polyphase suspensions that were broadly at internal equilibrium[22,71]. The common occurrence of granular harzburgites (i.e., olivine–orthopyroxene cumulate rocks) and pyroxene–chromite cumulates are explicitly forbidden during fractional crystallization by the peritectic relations among olivine, orthopyroxene, and chromite but are entirely consistent with equilibrium phase relations.

As our modeling of the Upper and Upper Main Zone indicates, our goal here is not to argue that AFC and fractional crystallization alone are not valid petrogenetic processes, but instead to demonstrate that the idealized ABC concept represents a process sufficient to account for much of the spectrum of rock types observed in the world's premier layered mafic intrusion, especially those world-class mineral deposits that are hosted by ultramafic macrolayers, and therefore cannot be ignored.

Furthermore, in those cases where cumulates of contaminated magmas display modal proportions departing from expected cotectic proportions, some form of ABC must be accepted as

having occurred. Indeed, a batch process is directly implied by several previous assertions that emplacement of the basal series dunites and harzburgites[29], Critical Zone chromite-bearing pyroxenites[43], and Main Zone gabbronorites[24] must have involved deposition of thick mushy layers, but these previous studies did not explore the implication that their mushy emplacement models are fundamentally inconsistent with AFC processes.

The implications for the mechanism of formation of layered mafic intrusions by injection of crystal mushes are far-reaching because the emplacement of each batch of magma, to form each macrolayer, is entirely independent from all of the other batches. Even in the Peridotite Zone of the Stillwater Complex, which has been regarded as the type locality for cyclic units representing the idealized products of fractional crystallization, no evidence can be found for genuine cyclicity due to fractional crystallization processes[22]. Rather, the compositions, textures, and modal variations in the Peridotite zone unequivocally require that a mixture of different mineral phases in cotectic proportions was mechanically sorted into layers including some that were nearly monomineralic within each larger macrolayer[22]. The ABC process does not require a large magma chamber to explain the sequence of rock types or to account for the lack of cyclicity; neither is the hypothetical existence of a magma chamber denied. We can consider layered mafic intrusions as products of numerous separate intrusive events in a long-lived magma column dominated by mushy zones punctuated by rare events when liquid-dominated magmas are transported and emplaced[5–7,55]. Macrolayers do not need to have formed in a younging sequence from bottom to top, although they might have. The hypothesis of mixing of fresh U-type magmas into resident A-type magmas of uncertain provenance to account for the sharp reversals in mineral assemblages that are associated with the major deposits of Cr and PGE[4,32–35] is not necessary, and the apparently random sequence of mafic and ultramafic layers in the Upper Critical Zone can be regarded as the consequence of injection of crystal-rich magma batches that experienced different paths through the lithosphere either (a) in the observed stratigraphic sequence or (b) out of stratigraphic sequence—either scenario is consistent with the observed occurrence of alternating mafic and ultramafic layers. Emplacement of one mushy macrolayer into still-hot older cumulates that may or may not remain partially molten need not produce easily recognizable chilled margins[12,71]. On the other hand, the level at which these crystal mushes are emplaced and ponded is mostly but not exclusively sensitive to the depth in a crustal column at which they achieve neutral buoyancy. Notwithstanding the effects of local stress fields[12], higher density ultramafic mixtures are likely to be emplaced beneath less-dense mafic zones, resulting in the general bottom-to-top sequence: ultramafic Lower Zone, pyroxenitic Lower Critical Zone, noritic Upper Critical Zone, gabbronoritic Main Zone, and noritic/dioritic Upper Zone below the pre-existing low-density volcaniclastic Rooiberg Group (Fig. 2). The operation of this density filter leads broadly to the overall trends of vertical changes in mineral compositions, e.g., Mg# in mafic minerals and An% in plagioclase that superficially resemble the results of fractionation within individual magmatic lineages even if the denser ultramafic layers are in some cases younger than the mafic rocks above them[12,13]. However, many complicated and apparently stochastic reversals in mineral compositions are common in detailed profiles[24,43,46], which can be attributed to the disordered emplacements of mushy macrolayers from similar but temporally discrete magmatic lineages, especially given that their emplacements are not determined by neutral buoyancy alone[70].

A profound implication for ore genesis is that chromitite and sulfide reef deposits in macrolayers of the RLS may each represent one relatively small batch of magma. The komatiitic parental

magma for the RLS contained a relatively high $Cr_2O_3$[43], and the upper-crustal ABC model proposes that the solid phases from the modeled pyroxenitic mushy Critical Zone include ~2% chromite. In 100 m of pyroxenite we might therefore expect to see a total of 2 m of chromitite, perhaps dispersed among several thinner layers, a proportion roughly in accordance with what is observed. Excess amounts of dense chromite could result from its preferential deposition while magma still laden with lighter silicate crystals passed overhead and exited the system.

Traditional models for deposition of stratiform PGE-rich sulfides depend on mixing within a large deep magma chamber of resident A-type magma with a new injection of PGE-rich sulfide-undersaturated U-type magma[4,33]. The model is fundamentally dependent on the existence of a magma chamber. However crustal assimilation by komatiite also easily triggers the segregation of sulfides, which scavenge PGE from hosting magma during vigorous transport and emplacement; a sheet of the B1 U-type magma as little as 250-m-thick contains sufficient PGE to account for the composition and grade-thickness of the Merensky Reef[50] if it has assimilated enough crustal material to attain a small degree of sulfide oversaturation. Once a sulfide-bearing, possibly chromite-rich crystal slurry is emplaced within the RLS and deflected into a sill-like body, lateral flow of the mush rapidly drives viscous segregation of minerals in which almost all of denser sulfide and/or chromite grains are deposited at the base. Where chromite is abundant, it can form a near-monomineralic chromitite layer above a thermally eroded hard substrate, and simultaneously the flotation of less-dense plagioclase at the top of the sill body can produce norite or anorthosites[65]. The chromitite layer could be further thickened due to subsequent multiple injections before its consolidation[65]. The lateral transport of dense slurries transgressively eroded their footwall rocks, accounting for the formation of potholes and regional magmatic unconformities[24]. Subsequent compaction[5] and annealing, coupled with percolation of residual liquid in the post-cumulate stage[6], may have persisted for hundreds of thousands of years below the zircon closure temperature[14]. These late processes could further enhance the development of annealed and compacted monomineralic layers lacking intercumulus melt[22,24], and also drive local isotopic disequilibrium[15,46,59,63].

The same type of contaminated and crystal-rich magma batches we envision in the formation of macrolayers in the RLS could equally well have formed smaller and less regularly-shaped intrusions containing high-grade chromitites like those of the Ring of Fire, Ontario, Sukinda, India, or Kemi, Finland or PGE-rich sulfide deposits like the Lac Des Iles Pd deposit of Ontario that occur in intrusions with irregular morphology but rock types effectively identical to those observed in the RLS[21,72]. Recognition that the processes involved do not require the existence of a magma chamber would open many new areas to exploration for these deposits of critically important strategic metals, areas previously overlooked purely because they do not contain vast layered mafic intrusions like the RLS. In contrast to the classical paradigm of fractional crystallization within liquid-dominated magma chambers, the state of the RLS as it grew may be better understood as a thick mushy reservoir containing transient pooled melt pockets, built by transcrustal assimilation processes including ABC and AFC, concepts which can be extrapolated to other large layered intrusions such as the Stillwater Complex[22] and to smaller irregularly shaped intrusions anywhere in the world.

## Methods

Isenthalpic assimilation simulations were carried out using the AlphaMELTS software, version 1.9, and more information about these thermodynamic models can be found in the Supplementary Tables where we provide copies of the melts files and environment files used. Model results appear in the Supplementary Data 1 file. We collected the major oxide contents of country rocks (Supplementary Fig. 3) and the closest exposures of middle crust (Supplementary Fig. 4) beneath the RLS, and identified potential representatives of the complex lithological association in this region (Supplementary Table 3). The estimated composition of granulite terrains from the interior of the North China Craton was considered to be representative of lower cratonic continental crust in general and used as the assimilant at lower crust levels[57]. Based on similarities in major oxides, the trace-element compositions of upper- and middle-crustal contaminants (Supplementary Table 3) were mostly assumed according to the global average values of upper continental crust[73] and Archean gray gneisses[74], respectively. Initial enthalpies and phase assemblages of these crustal assimilants were estimated under the suitable pressure-temperature conditions (upper crust, 0.2 GPa, 300 °C; middle crust, 0.45 GPa, 390 °C; lower crust, 1 GPa, 770 °C). In alphaMELTS, crustal material is incrementally added to the system, and an isenthalpic calculation employs entropy maximization to solve for thermodynamic equilibrium between silicate liquid and solid phases at constant pressure. Any resultant crystals can be equilibrated with or discarded from residual liquid (ABC or AFC, respectively), and the remaining system becomes the starting point for the next increment of assimilation. Following wholesale crustal assimilation at middle (0.45 GPa) and lower crust levels (1 GPa), further cooling of ascending magmas in the conduit was represented by an isobaric crystallization at 0.2 GPa and fayalite–magnetite–quartz solid oxygen buffer.

Variable rare-earth element partition coefficients of clinopyroxene/melt and feldspar/melt were calculated based on lattice strain theory, while constant partition coefficients for remaining elements and melt-solid pairs were taken from a comprehensive review[54]. Because slight differences in the Gibbs free energy among various candidate model pyroxenes may confuse the choice of the second pyroxene after orthopyroxene, we corrected the pyroxene/melt partition coefficients when the algorithm improperly terms a high-Ca clinopyroxene as the second orthopyroxene in the B3-Gn (Main Zone) lineage.

Based on evolution model of depleted mantle at 2055 Ma, the $(^{87}Sr/^{86}Sr)_i$ and $\varepsilon_{Nd}$ of mantle-derived magma are calculated as 0.7016 and 3.97, respectively. The $\delta^{18}O$ of an uncontaminated mantle-derived magma is widely considered as 5.7‰ in light of the global range of 5.4–6.0‰ in fresh MORB glasses[60]. $(^{87}Sr/^{86}Sr)_i$ and $\varepsilon_{Nd}$ values of middle crust and lower crust are the averages of extensive collected data in this region (Supplementary Table 4). Limited to rare data of sedimentary Pretoria Group, the O isotopic data of upper crust are assumed mainly with reference to the overlying Rooiberg Group and adjacent Dolomite.

Pretoria Group rocks have the higher $\delta^{18}O$ (~9–15‰) than the averages of overlying volcanites (7.36‰), granophyres (6.6‰), and granites (7.35‰), and a moderate value (9.6‰) was set for the upper-crustal materials (Supplementary Table 4). Average $\delta^{18}O$ values of the Vredefort Dome and Limpopo belt mostly fall in the range 9–10‰, but we adopted a slightly larger $\delta^{18}O$ for middle crust assimilant (10.6‰) assuming a possible greater contribution from $\delta^{18}O$-enriched metapelites. The proposed lower crust has moderate $Mg^\#$ and $SiO_2/Al_2O_3$ ratios, corresponding to the features of intermediate-type lower-crustal granulite xenoliths that has an average $\delta^{18}O$ of 9.2‰[75].

## Data availability

The authors declare that all relevant data necessary to reproduce the figures presented in this paper are available within the article and its supplementary information files. A Supplementary Information file provides tables and figures in support of the main text as well as copies of the alphaMELTS melts files and environment files. A Supplementary Data 1 file contains all the model results from alphaMELTS. Source data are provided with this paper.

## Code availability

AlphaMELTS is available for free download from the Caltech Magmasource website at https://magmasource.caltech.edu/alphamelts/ where support can be found for installation and execution of the software. Environment and input files and a summary of the steps taken to conduct the models using alphaMELTS are available in the Supplementary files for this article.

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

## Acknowledgements
J.E.M. acknowledges funding from NSERC Discovery Grant program.

## Author contributions
J.E.M., Z.Y., and M.C.J. developed the ideas and shared initial pilot-scale modeling efforts. Z.Y. and J.E.M. compiled literature and data sources. Z.Y. performed the modeling in depth with frequent inputs from J.E.M. and M.C.H. and produced all the figures and tables. J.E.M. and Z.S.Y. wrote the manuscript.

## Competing interests
The authors declare no competing interests.
