## [Peer Review File · Nature Communications]

REVIEWER COMMENTS

Reviewer #1 (Remarks to the Author):

Review of 'Formation of the Rustenburg Layered Suite by assimilation – batch crystallisation (ABC) and – fractional crystallisation (AFC)

Authors Yao et al.

This manuscript concerns the Rustenburg Layered Suite (RLS) – the layered mafic-ultramafic intrusion portion of the Bushveld Complex. The authors set up the hypothesis that assimilation-batch crystallisation (ABC), a variant on the more commonly invoked assimilation-fractional crystallisation (AFC) mechanism in crystallising magmatic systems, may have played an important role in forming of the RLS. The topic and approach employed are significant and timely for a number of reasons. One is that the study offers a reasonable alternative to the common paradigm of AFC, and shows that it might actually work for the Bushveld. Another is the general importance of understanding crystallisation of the RLS with respect to the unrivalled base and precious metal resources it contains. Finally, the Bushveld and other layered intrusions are currently at the centre of an important debate over the way in which basaltic magmatic systems solidify. Broadly speaking, this may be from largely crystal-free 'chambers' of magma in the classic sense, or as iteratively-constructed mushy zones that may never have that much melt present at any one time.

There is a lot to like about this manuscript. Overall it is well presented and topical. The modelling seems robust and contextual aspects/implications of the study all seem well laid out. I have some thoughts and comments below for the authors that ultimately I don't think will constitute anything more than minor-moderate revisions, after which I think the work is eminently suitable for publication in Nature Communications.

I think the abstract could be a bit more engaging. Perhaps leave some of the abbreviations and jargon for the main text, and just focus on summarising the important processes in a more accessible way? The mineral deposits implication at the end is an important one to be sure, but perhaps a more important one is another nail in the coffin for the view that all layered intrusions formed from melt dominated magma chambers.

Intro and early part of discussion (eg P17): I think a little more clarity around where the assimilation is going on would be good. There is a danger that (as you point out yourselves early on) the reader won't see the difference between ABC and AFC as anything more than academic. Put simply, to my mind there is ascent, where the magmas are transported from the mantle, and there is emplacement, where they are introduced to the chamber. Considering the ascent stage, there are some questions as to the degree to which the conduits within which ABC occurs are open systems and the extent to which they may 'mature' and thermally condition the host rock through which they flow. Assuming that the composition of the komatiite magma being extracted from the mantle source remains constant, then the degree to which it can assimilate wall rock or crustal material during ascent should vary (decrease) as the plumbing system matures? Now as a result of this insulating effect, we could get a situation where batches of quite primitive magma can repeatedly be emplaced into the mushy RLS pile, with lots of assimilation potential. Only what is being assimilated now are the wall rocks to the intrusion and cumulates themselves, possibly leading to repeated melt rock reaction events etc.

Following on from the point above, I felt the interplay of convection and flow during ABC as expressed here to be a little confusing. In simple terms, thinking about transport of magma in a sheet like conduit, especially where that viscosity is as low as suggested (ie komatiite) and where transport is as fast as suggested (m/s), won't unidirectional flow this strong inhibit convection? Can the differences between or roles of these processes be teased apart a little more? Presumably in the ascent stage, these conduits are vertical or subvertical – are there implications there for

limitations on convection? Or is the contention that at higher Rayleigh numbers, 'turbulent convection' (bottom of P10) and flow are the same?

When discussing the model results in the context of isotopic variations through the RLS sequence (eg around P15), I would be interested to hear a little more detail on how the authors believe the coupled Sr and Os data of Ronny Schoenberg (1999; EPSL) can be explained by ABC. Essentially they report almost ubiquitous isotopic disequilibrium within and between chromitite seams throughout the Critical Zone (Lower, Middle and Upper Groups). Sr isotopes come from interstitial plag, and Os isotopes are mainly from chromite (which probably means HSE-rich inclusions).

I take the point that it is an end member scenario for the sake of the modelling. But the notion expressed in places throughout the text that in ABC, entrained solids are deposited 'all at once' (e.g., Fig 1 caption) and that after 'dumping...as masses', 'some crystal sorting' (bottom of P17) arranges everything into the layers we see in the solidified products, leaves me with some unanswered questions. For example, if this mechanism is prevalent (as depicted in Fig 1b), why don't we see more refractory fragments of wall rock and other bits and pieces in layered cumulates? Cumulate autoliths are common in layered cumulates (usually 10s cm to metres in size) but often show signs of having deformed layering, so were introduced to layered cumulate later (eg exemplified in the Skaergaard, though admittedly that is not the best example here). In other words, if the authors are right, why is so much layered cumulate so lithologically/texturally homogeneous, including in the Bushveld, within respective layers?

Bottom of P19-P20: Ok, but there are still some really strong field observations for in situ crystallisation of chromitite seams (ie the Latypov JPet papers on Merensky and UG2), such as the way they line footwall topography (including overhangs) – accumulation cannot be by crystal settling here, so how do you reconcile this with your ideas? We see the same behaviour of chromitite seams on Rum, incidentally, which has been very important in shaping my views of chromitite petrogenesis.

Some other minor things:

Page 5, 2nd paragraph: Could maybe add some thicknesses of the different individual zones and subzones of the RLS.

Page 6, right at the top: So are the 'locally variable' cumulates referred to essentially laterally equivalent facies of the same recharge events?

Page 7, 4th para: The komatiite is the agent causing the assimilation, so it reads a little odd to me to say it undergoes assimilation. Could just say '...magma that assimilated the quartzitic floor...'

Page 8, 2nd para: I think your ref 45 only deals with 187Os/188Os, so I don't know where 186Os comes from here.

Page 9, bottom: I know it is in the Fig 3 caption to some extent, but I think it would be helpful here to add a sentence on what the 'key lithologies' are and why.

Page 10, Fig 3 caption. I think this needs a little work. It could do more to guide the reader into what the plots show, including the insets. Suggest one or two more sentences of higher order content description, then work down into the details, systematically. I don't think it is necessary by any means, but is there any point in showing some calculated compositions from AFC type processes on the plots to see where they bring us?

Page 10, second para: It is stated that two distinct scenarios are considered, but going through to Page 11, three are actually presented.

Page 11, close to top: 'This next stage...' You mean the gravity separation referred to previously? Not 100% clear. I like that the fate of the 'supernatant magma' can be resolved with the B1 marginal sills.

Page 13: The caption to Figure 4 is a bit sparse. Can a little detail for each panel be added? Could the symbols for same lithologies be made the same as Fig 3?

Page 18, second para: Maybe 'startling', but this is a little colloquial, suggest rephrase

Brian O'Driscoll
July 7th 2020

Reviewer #2 (Remarks to the Author):

Review of: Formation of the Rustenburg Layered Suite by assimilation – batch crystallization (ABC) and – fractional crystallization (AFC) by Zhuo-Sen Yao, James E. Mungall and M. Christopher Jenkins

Submitted to Nature Communications

The Rustenburg Layered Suite (RLS) forms a major part of the Bushveld Igneous Complex and is remarkable not just for its spectacular igneous layering, known to geologists worldwide, but also for its huge proven and future economic value. In particular, much of the world's platinum group element resource occurs within the RLS. The manuscript by Yao and others is a modelling effort aimed at understanding how the remarkable layering was formed in the RLS and what this might reveal about magma chamber processes. These authors argue, based on their modelling, that assimilation and batch crystallization (ABC), rather than assimilation and fractional crystallization (AFC), explain the composition of the majority of RLS rocks. The consequence of arguing for ABC rather than AFC may seem a little arcane, but it is important. This argument indicates that, instead of the textbook concept so well known to those recently introduced to geology, of a vast (and often balloon shaped!) magma chamber raining out crystals to form layers, that the RLS likely formed from discrete sills or regions of partial melt and crystals. Since the RLS forms such a major part of the Bushveld Igneous Complex, this reasoning removes the seemingly unreasonable requirement that this huge complex of igneous rocks was once entirely molten at the same time. Inevitably this conclusion might eventually provide some clarity as to how economically viable deposits, such as the Merensky Reef or G chromitite within the RLS were formed. The conclusion also falls in line with evidence for silicic systems for a partially molten 'magma chamber' and with recent arguments for much smaller layered intrusions having a similar origin.

Given the above statements, the work by Yao et al. is certainly of the impact and relevance for publication in Nature Communications, but I would suggest with moderate revision.

In general, the modelling presented is well presented, and I have only a few comments on this aspect of the work. However, I note that this is a model, and that the endmember parameters and assumptions of depth of crystallization, geotherm etc... are what drive the model. What really needs revision is the explanation of the importance of the work and discussion of some of the assumptions, as well as of supporting observations that might also indicate that the RLS was never fully molten at any given stage. To hopefully aid the authors with their revision, I have general suggested comments, and inline comments. Overall, I think that this is a useful addition to the petrological literature, and opens some intriguing new questions as well as perhaps helping to close the textbook on perhaps more ingrained and thermodynamically poorly constrained ideas that have persisted for decades, or at least to encourage more lateral thinking.

General Comments

1. In general, I think the authors could make the paper a little bit more accessible to readers while also helping to address the 'arcane' comment above. This could start with the title, which might be impenetrable to some, and less than exciting to others. I would suggest something along the lines of "Formation of layering in the world's largest igneous intrusion through small-scale melt crystallization" or something of that nature. I leave it to the author's as to the exact terminology, but I suspect they might agree that the current title, while factual, is not necessarily going to have researchers or Bushveld protagonists scrambling to read it. On the same lines, I have specific comments about removing the criticism of being in danger of becoming arcane in the inline comments on the PDF.

2. It would be useful if some more thermal arguments that might support their assessment are brought into the discussion. For example, the thermal aureole of the RLS is remarkably limited given that it was supposed to have been an entirely molten magma chamber at one stage. Equally, the preservation of a 'chilled margin' with komatiitic affinity is remarkable if the RLS was indeed completely molten, or even hot, as this model also argues. The authors might also wish to show more explicitly how an AFC model fits or does not fit with the lower most portion of the RLS – this is shown in the figure, but a brief clearer mention to it would help.

3. The authors could expand on describing the chill margin with spinifex olivine and why this is a good proxy for a melt composition. If the model of a vast molten magma chamber with the same starting composition were true, then this would be reasonable. However, we are not discussing that type of a model. We are discussing a pulsed addition of melt into a complex network of crystal mush or something akin to crystal mush (i.e., in many places 99% crystal). How does one preserve a spinifex texture with 7-9 km of igneous material above that must have at least been fairly hot (maybe not hot enough to be molten, but hot). How is this preserved given the argument for the ease with which komatiite becomes contaminated? Ultimately, much of this is important, as turbulent flow is invoked at the high differential temperatures of the komatiite liquid to the crustal assimilated. This may be true at the inception of the RLS, but it cannot be true during the later phases of layering when the system has 'warmed up' and I imagine that's part of the logic requirement for polybaric crystallization as the modelling progresses.

4. The authors point out that "a hypothetical dimensionless thermodynamic black box ABC process" has been successful. This raises the point that, if the authors presented their model, perhaps as an excel spreadsheet, or maybe whatever program format they used (MELTS output), this would really be helpful to the community and would be an important contribution from the work. Can the authors do this?

5. The modelling of isotopic compositions in Fig. 6 is not well expressed. Are these simply mixing models, or do they incorporate information from the MELTS modeling? The bars for the different zones are bit complicated. In general, the authors need to explain the modeling in this figure much more clearly. In comments on the supplement PDF, I also have queries about the endmember compositions used. Using the North China Craton composition seems really odd here, given that the geographic location is very different. Are there no lower crustal xenoliths from local kimberlites of the Kaapvaal or environs that could be used instead?

6. The work could be more explicit about the broad zonal classification being oversimplified for its regional-scale utility, while the cumulate layers show numerous and complex mesoscale variations in their spatial distribution. This is a nuance that some might not catch, while simultaneously using the chromitites as markers. It is an important point, difficult to explain by a vast AFC magma chamber model, but perhaps also equally difficult to explain with an ABC model as well. For example, the presence of potholes and vast accumulations of chromitite is not explicitly modelled here.

7. This brings me to the greatest area of concern surrounding the manuscript, which is the development of monomineralic layers and sulfide behavior. The formation of monomineralic layers is never expressly given in the paper, nor does the modelling ever approach a 'pure magnetite' or a 'pure chromitite'. This is critical, as these layers are stratiform and span nearly the entire complex. What's more, these layers have been the subject of intense scientific scrutiny and numerous hypotheses, including an insitu origin, have been given for them. I cannot see how 'small scale melt batches', as expressed in the last paragraph of the paper, could ever achieve these accumulations, unless small scale is really quite big, and merely a relative term. These chromitites exist throughout much of the lower portion of the sequence. Magnetite and Nelsonite layers occur in the upper portions and these require explanation too. The MELTS modelling cannot achieve this, yet a major conclusion is that ABC can generate such layers, as well as the S-rich layers forming features like the Merensky Reef. I don't think this paper gets close to addressing these major features of the RLS. Presumably, the authors leave open the possibility of nearly every idea, other than formation of these layers by crystal settling from a big open and molten magma chamber. This is certainly a weak point in the paper, and will be exploited, if the authors do not 'plug the hole' here by perhaps providing some explicit examples of how the modelling relates to such features. Alternatively, they might simply wish to admit that the vast accumulation of Cr spinel and magnetite require further work in the context of the RLS.

Inline comments:

Because no line numbers were given, I provide inline comments on both the main manuscript and supplement PDFs – attached.

Reviewer #3 (Remarks to the Author):

It is an excellent paper that makes a convincing (and plausible argument) against the RLS (at least in its economically important (roughly) lower half) being derived by the solidification of a melt-dominated chamber. Instead, the authors argue that this part of the RLS formed from the emplacement of non-sequentially injected crystal slurries. These slurries were processed in deeper storage chambers – where they underwent differentiation by ABC (Assimilation Batch Crystallization) – prior to emplacement in the RLS. The economically important Upper Critical Zone, specifically, underwent two stages of ABC with emplacement of its feldspathic units first, followed by its pyroxenites. In contrast, the upperparts of the Main Zone and Upper Zone (termed the UUMZ) were derived by 'conventional' AFC of large volumes of ferrobasaltic magma in the chamber. The quantitative modelling that the authors use in support of their model appears to be sound and it can be reproduced using the supplementary information (though admittedly I did not have time to do this). The case the authors make is important because there is a longstanding convention (that is still perpetuated now) that the RLS is the product of crystal sorting and melt fractionation (+ recently by in situ crystallization) in a melt-dominated system. However, there is abundant (and growing) field and chemical evidence that this is not the case, and that magmas emplaced into the RLS were processed in deeper magma reservoirs and they entrained crystal cargoes during emplacement (in accord with the magma plumbing dynamics of other continental flood basalt provinces throughout geological time).

I splattered some notes in the attached pdf of the manuscript for the authors to look at. They mostly relate to the following general points and only seek to help the authors strengthen their arguments:

- The connection between the non-sequential assembly model for the Critical Zone and its economically important layers (PGE reefs + chromitites) needs to be better developed. For instance, the final sentence of the Abstract comes out of the blue and there is certainly a paragraph missing in the Discussion that should explain the following points; how are the formation of PGE reefs in the RLS explained by the melt-dominated chamber model, and how can the authors preferred model better explain their formation? It would be good to see a clear

hypothesis put forward in this regard that can be thought about and tested in the future.

- I made a few comments in places that refer to the research that Kruger presented regarding a plethora of papers in the 1980s/early 90s using initial $87\text{Sr}/86\text{Sr}$ ratios. Kruger (1994 and 2005) summarised these data to argue that the RLS formed from two stages (Integration and Differentiation Stages). I do not think the authors have done this work justice in the relevant parts of the manuscript – and, if one was being highly critical, the authors model could just be considered as a more detailed explanation of Kruger's work. I think the important point the authors need to make is that Kruger perhaps considered his model in the context of a melt-dominated chamber - with little detail about komatiitic melts processing crust and generating isotopically modified slurries – maybe the authors did make this point in the article, but I did not pick it up?

- I do not think there was any mention of either inter- and intra-crystal isotopic disequilibrium that has been documented in the RLS (see data in Prevec et al. 2005; Chutas et al. 2012; Roelofse & Ashwal, 2012; and lots of other recent papers using laser Sr isotope data in plagioclase). It would be good to see these data considered at some point and how it could be explained in the context of a transcrustal magmatic system. They might also want to consider mentioning the role that intercumulus liquid percolation played in the isotopic signatures of the cumulates (+ minerals) as they are in favour of the existence of thick piles of mush during RLS solidification.

- This may be beyond the scope of the present article, but it is perhaps something to consider: The discrepancy between zircon Hf isotopes (constant throughout the RLS; after Zirakparvar et al. 2014) and the (?) common presence of isotopic disequilibrium in the major cumulate-forming silicate phases (many solution and laser papers on this) is puzzling. Is it possible that the homogeneous Hf isotope signature of zircon is a product of a isotopically homogeneous carrier melt that entrained ABC processed slurries from depth? Or could this homogeneous Hf signature be explained by RLS-level contamination (recent Zeh paper in JPet)? We've been thinking about this problem here in Joburg and we would be happy to see the authors present a solution to this problem in the context of their model.

- According to mineral compositional profiles for the entire RLS (e.g., in Cawthorn, 2015) – it appears that pyroxenites of the UCZ have lower Mg# (in opx or ol) compared to parts of the LCZ or LZ. Assuming (here comes arguably another misplaced convention of the RLS...) that there is a smooth upward decrease in the Mg# of ferromagnesian minerals in sub-UUMZ parts of the RLS, then how do the authors reconcile this with their non-sequential emplacement model? Would the mineral compositional profiles not look a bit more random in this part of the RLS? It would be useful to see this addressed in the discussion.

Ben Hayes

Johannesburg, 3rd August, 2020

General comments: We are grateful to the reviewers for their encouraging take on our manuscript and for the many suggestions they have made to improve it. In some cases their comments seem to drift into discussion of points that we feel are very interesting but impossible to address in the present manuscript, so in some cases we respond with our own discussion but do not make major adjustments to the text. We have highlighted all revisions in yellow and provide some responses below.

REVIEWER COMMENTS

Reviewer #1 (Remarks to the Author):

Review of 'Formation of the Rustenburg Layered Suite by assimilation – batch crystallisation (ABC) and – fractional crystallisation (AFC)

Authors Yao et al.

This manuscript concerns the Rustenburg Layered Suite (RLS) – the layered mafic-ultramafic intrusion portion of the Bushveld Complex. The authors set up the hypothesis that

assimilation-batch crystallisation (ABC), a variant on the more commonly invoked assimilation-fractional crystallisation (AFC) mechanism in crystallising magmatic systems, may have played an important role in forming of the RLS. The topic and approach employed are significant and timely for a number of reasons. One is that the study offers a reasonable alternative to the common paradigm of AFC, and shows that it might actually work for the Bushveld. Another is the general importance of understanding crystallisation of the RLS with respect to the unrivalled base and precious metal resources it contains. Finally, the Bushveld and other layered intrusions are currently at the centre of an important debate over the way in which basaltic magmatic systems solidify. Broadly speaking, this may be from largely crystal-free 'chambers' of magma in the classic sense, or as iteratively-constructed mushy zones that may never have that much melt present at any one time.

There is a lot to like about this manuscript. Overall it is well presented and topical. The modelling seems robust and contextual aspects/implications of the study all seem well laid out. I have some thoughts and comments below for the authors that ultimately I don't think will constitute anything more than minor-moderate revisions, after which I think the work is eminently suitable for publication in Nature Communications.

I think the abstract could be a bit more engaging. Perhaps leave some of the abbreviations and jargon for the main text, and just focus on summarising the important processes in a more accessible way? The mineral deposits implication at the end is an important one to be sure, but perhaps a more important one is another nail in the coffin for the view that all layered intrusions formed from melt dominated magma chambers.

We have modified the abstract, removed some abbreviations and highlighted that the crystal-dominated mush and associated slurries during emplacement of the layered RLS may largely obviate the need to consider a classic magma chamber.

Intro and early part of discussion (eg P17): I think a little more clarity around where the assimilation is going on would be good. There is a danger that (as you point out yourselves early on) the reader won't see the difference between ABC and AFC as anything more than academic. Put simply, to my mind there is ascent, where the magmas are transported from the mantle, and there is emplacement, where they are introduced to the chamber. Considering the ascent stage, there are some questions as to the degree to which the conduits within which ABC occurs are open systems and the extent to which they may 'mature' and thermally condition the host rock through which they flow. Assuming that the composition of the komatiite magma being extracted from the mantle source remains constant, then the degree to which it can assimilate wall rock or crustal material during ascent should vary (decrease) as the plumbing system matures? Now as a result of this insulating effect, we could get a situation where batches of quite primitive magma can repeatedly be emplaced into the mushy RLS pile, with lots of assimilation potential. Only what is being assimilated now are the wall rocks to the intrusion and cumulates themselves, possibly leading to repeated melt rock reaction events etc.

The issues of time-dependent 'maturation' of the feeder system are interesting but far too complex and poorly constrained to address here. We do consider the effects of thermal regime on the style of assimilation, from dynamic and rapid ABC in the shallow crust to slow and more homogenized AFC in the hot lower crust, but we do not feel confident enough to connect these thermal regimes to any temporal evolution apart from a mention that was already present in the original text that by the time the UUMZ formed, the lower crust might have undergone considerable heating by the previous magmatic activity.

Following on from the point above, I felt the interplay of convection and flow during ABC as expressed here to be a little confusing. In simple terms, thinking about transport of magma in a sheet like conduit, especially where that viscosity is as low as suggested (ie komatiite) and where transport is as fast as suggested (m/s), wont unidirectional flow this strong inhibit convection? Can the differences between or roles of these processes be teased apart a little more? Presumably in the ascent stage, these conduits are vertical or subvertical – are there implications there for limitations on convection? Or is the contention that at higher Rayleigh numbers, ‘turbulent convection’ (bottom of P10) and flow are the same?

Good suggestion! We have tried to make the dynamic convection clearer for the readers in the revised manuscript. An entire paper could be written on this subject and we cannot go into depth in the present short format article, but we can comment to the editor and the reviewer that turbulent flow is characterized as an interacting cascade of eddies over a range of length scales (for the large scale that may have the similar length scale to the width of dyke; to the small scales that may equal to the size of minerals). Because the turbulent flow is highly disordered in the time-space domain, it is still difficult to describe completely even with the help of fluid dynamic modelling. Hence, the Reynolds number, a dimensionless value that is defined as the ratio of momentum force to viscous force, is widely used to point out a turbulent flow regime where the momentum force outweighs the viscous force. Different flows with higher Reynolds numbers have the similar dynamic features, no matter whether it is in a vertical dyke or horizontal sill (because the convection and eddies are also common in a smaller length scale). On the other hand, for the ascending magma in a vertical conduit, the gravitational relative settling velocity of minerals are less than the ascending rate of magma flow in most cases, which suggests that the crystal grains ascend with the magma flow and may remain in equilibrium in a prolonged time, even in the absence of turbulent flow. We expect to publish more work elaborating on these themes in more specialist journals in the near future.

When discussing the model results in the context of isotopic variations through the RLS sequence (eg around P15), I would be interested to hear a little more detail on how the authors believe the coupled Sr and Os data of Ronny Schoenberg (1999; EPSL) can be explained by ABC. Essentially they report almost ubiquitous isotopic disequilibrium within and between chromitite seams throughout the Critical Zone (Lower, Middle and Upper Groups). Sr isotopes come from interstitial plagioclase, and Os isotopes are mainly from chromitite (which probably means HSE-rich inclusions).

The Schoenberg et al., (1999) Re-Os isotopic data in CZ give a Re-Os isochron age (2043 ± 11 Ma) that differs from the high-precision crystallization age of the RLS is about 2055 ± 1 Ma; therefore post-cumulus processes may have reset the Re-Os isotope system. On the other hand, the interstitial plagioclases in the LZ and CZ are highly zoned, and their in-situ Sr isotopic data mostly remain in disequilibrium with the coexisting minerals. Here, we tend to support the new explain of Hepworth et al. (2020) for the formation of similar scenes in Rum layered intrusions, in which the highly localized reactive melt percolation drives the isotopic heterogeneity. In the revised manuscript, we have mentioned this. Perhaps more importantly, we also point out that the major element behaviour of a system undergoing rapid and dynamic assimilation and transport may lead to fairly well-equilibrated mineral compositions and modes while leaving locally derived isotopic variations within crystals prone to disequilibrium. Assimilation of amphibolite and granite by komatiite will produce a very

similar cumulate and contaminated liquid but one granite may have quite different Sr isotopic character depending on its age, so the batches of crystals so formed by assimilation at different points along the transport path may have similar major element compositions but radically different Sr isotope ratios. If the editor thinks we need to belabor this point more in the text we would be happy to, but we are concerned about word length.

I take the point that it is an end member scenario for the sake of the modelling. But the notion expressed in places throughout the text that in ABC, entrained solids are deposited 'all at once' (e.g., Fig 1 caption) and that after 'dumping...as masses', 'some crystal sorting' (bottom of P17) arranges everything into the layers we see in the solidified products, leaves me with some unanswered questions. For example, if this mechanism is prevalent (as depicted in Fig 1b), why don't we see more refractory fragments of wall rock and other bits and pieces in layered cumulates? Cumulate autoliths are common in layered cumulates (usually 10s cm to metres in size) but often show signs of having deformed layering, so were introduced to layered cumulate later (eg exemplified in the Skaergaard, though admittedly that is not the best example here). In other words, if the authors are right, why is so much layered cumulate so lithologically/texturally homogeneous, including in the Bushveld, within respective layers?

First of all, crustal xenoliths can be easily found in the Marginal Zone, the contact sequence between the LZ and CZ, the top of UZ, and also the chromitite layers (Maier et al., 2013). On the other hand, the komatiitic parental magmas can easily melt crustal xenoliths, and the time-scale for this process is about several minutes (Robertson et al., 2015 EG paper), so they may be expected to be relatively rare in the main body of the RLS. Hence, the crustal xenoliths in the komatiite flows are difficult to be preserved, and only may be found in the chilled rocks with a quick cooling, e.g., the Marginal Zone. Cognate xenoliths (autoliths) derived from the entire crustal column are common throughout the RLS, including the UMZ (e.g., Bourdeau PhD thesis Wits 2020). The cumulate autoliths can be the products of the injection and emplacement of crystal-dominated slurries into a semi-consolidated cumulate. At the same time, the transfer of this semi-consolidated crystal slurry along sill floors can quickly drive the viscous particle segregation due to the contrasts of densities and grain sizes for different phases. The cumulate layers will be progressively sorted during lateral transport, leading to the pronounced layering of chromitite, pyroxenite, norite and anorthosite layers. Subsequent expulsion and deformation can also take place in the final deposits. This dynamic process has been demonstrated by the recent analog experiments (Forien et al., 2015 JPet), and can account for the lithologically/texturally homogeneous macrolayers. We also cited a figure from this analog experiment in the following response. To emphasize this viewpoint we have mentioned the role of layer formation by crystal sorting in transported slurries several times in the revised manuscript.

Bottom of P19-P20: Ok, but there are still some really strong field observations for in situ crystallisation of chromitite seams (ie the Latypov JPet papers on Merensky and UG2), such as the way they line footwall topography (including overhangs) – accumulation cannot be by crystal settling here, so how do you reconcile this with your ideas? We see the same behaviour of chromitite seams on Rum, incidentally, which has been very important in shaping my views of chromitite petrogenesis.

The chromitite seams in the Unit-10 of Rum layered intrusion are mostly dispersed in the intercumulus spaces among plagioclase crystals (Hepworth et al., 2020), while the Merensky chromite occurs lining an obvious erosive unconformity, and UG2 shows unmistakable

signs of mushy interaction with underlying anorthositic mush (e.g., flame structures and domes of anorthosite interfingering with the chromitite). The three examples are different and only the UG2 is consistent with the mechanisms we describe here. We are happy to consider the Rum example to have resulted from reactive transport and the Merensky example to have resulted from reactions between Cr-rich ultramafic magma and its eroded mafic substrate. It is really beyond the scope of the present article to attempt to cover the details of the origins of every type of chromite seam. Furthermore, Latypov et al. (2016, Nature Communications) abandoned the in-situ crystallization model only one year later, and proposed a new model that the chromitite layers are formed by the replenishments of chromite-only-saturated melt from the deep-seated reservoirs....

In our opinions, the monomineralic chromitite layers in RLS can be formed by the viscous particle segregation during the lateral transport of semi-consolidated crystal slurries, which has solid support from the recent analog experiments from Forien et al. (2015, JPet), and reconciles with our ABC model here. We have explained this in the new revised manuscript while trying not to allow ourselves to be diverted into attempting to explain every feature of layered intrusions. The main focus here is on the means by which mixtures of crystals could form, arrive and organize themselves into layers.

Some other minor things:

Page 5, 2nd paragraph: Could maybe add some thicknesses of the different individual zones and subzones of the RLS.

We have added the thicknesses of the different individual zones of the RLS.

Page 6, right at the top: So are the 'locally variable' cumulates referred to essentially laterally equivalent facies of the same recharge events?

We have added some wording to the effect that the footwall and hanging wall sequences of the chromitites are extremely variable from place to place, and we suggest that this argues against any model for the chromitite belonging to a predictable crystallization sequence; instead it is easier to accommodate in a model of emplacement of the chromitite-bearing macrolayers into pre-existing cumulates of various descriptions.

Page 7, 4th para: The komatiite is the agent causing the assimilation, so it reads a little odd to me to say it undergoes assimilation. Could just say '...magma that assimilated the quartzitic floor...'

We have changed based on your suggestion.

Page 8, 2nd para: I think your ref 45 only deals with 187Os/188Os, so I don't know where 186Os comes from here.

We have changed this.

Page 9, bottom: I know it is in the Fig 3 caption to some extent, but I think it would be helpful here to add a sentence on what the 'key lithologies' are and why.

We have added some explanations for the key lithologies here.

Page 10, Fig 3 caption. I think this needs a little work. It could do more to guide the reader into what the plots show, including the insets. Suggest one or two more sentences of higher order content description, then work down into the details, systematically. I don't think it is necessary by any means, but is there any point in showing some calculated compositions from AFC type processes on the plots to see where they bring us?

We have added one sentence to give a high order content description for the Figure 3. The proposed parental magma for the UUMZ is generated by the AFC model in the lower crust, and is shown as the pentagon in Figure 3. Because the ultramafic-mafic grains are constantly removed from the systems during the AFC, the remaining melt has a far lower MgO content relative to the cumulates.

Page 10, second para: It is stated that two distinct scenarios are considered, but going through to Page 11, three are actually presented.

We say that there are two distinct varieties of ABC model and a third model of AFC, so in fact we have mentioned all three.

Page 11, close to top: 'This next stage...' You mean the gravity separation referred to previously? Not 100% clear. I like that the fate of the 'supernatent magma' can be resolved with the B1 marginal sills.

We have reworded this section to clarify the two stage ascent and emplacement process.

Page 13: The caption to Figure 4 is a bit sparse. Can a little detail for each panel be added? Could the symbols for same lithologies be made the same as Fig 3?

We have added the details in the caption of Figure 4. The same symbols were firstly adopted for the Figure 4, but it is difficult to distinguish the measured data and modelled results when the B1-3 magmas are represented by the colorful crosses that are the same with Fig. 3. Hence, we changed the symbols in figure 4.

Page 18, second para: Maybe 'startling', but this is a little colloquial, suggest rephrase

Removed it.

Brian O'Driscoll
July 7th 2020

Reviewer #2 (Remarks to the Author):

Review of: Formation of the Rustenburg Layered Suite by assimilation – batch crystallization (ABC) and – fractional crystallization (AFC) by Zhuo-Sen Yao, James E. Mungall and M. Christopher Jenkins

Submitted to Nature Communications

The Rustenberg Layered Suite (RLS) forms a major part of the Bushveld Igneous Complex and is remarkable not just for its spectacular igneous layering, known to geologists worldwide, but also for its huge proven and future economic value. In particular, much of the world's platinum group element resource occurs within the RLS. The manuscript by Yao and others is a modelling effort aimed at understanding how the remarkable layering was formed in the RLS and what this might reveal about magma chamber processes. These authors argue, based on their modelling, that assimilation and batch crystallization (ABC), rather than assimilation and fractional crystallization (AFC), explain the composition of the majority of RLS rocks. The consequence of arguing for ABC rather than AFC may seem a little arcane, but it is important. This argument indicates that, instead of the textbook concept so well known to those recently introduced to geology, of a vast (and often balloon shaped!) magma chamber raining out crystals to form layers, that the RLS likely formed from discrete sills or regions of partial melt and crystals. Since the RLS forms such a major part of the Bushveld Igneous Complex, this reasoning removes the seemingly unreasonable requirement that this huge complex of igneous rocks was once entirely molten at the same time. Inevitably this conclusion might eventually provide some clarity as to how economically viable deposits, such as the Merensky Reef or G chromitite within the RLS were formed. The conclusion also falls in line with evidence for silicic systems for a partially molten 'magma chamber' and with recent arguments for much smaller layered intrusions having a similar origin.

Given the above statements, the work by Yao et al. is certainly of the impact and relevance for publication in Nature Communications, but I would suggest with moderate revision.

In general, the modelling presented is well presented, and I have only a few comments on this aspect of the work. However, I note that this is a model, and that the endmember parameters and assumptions of depth of crystallization, geotherm etc... are what drive the model. What really needs revision is the explanation of the importance of the work and discussion of some of the assumptions, as well as of supporting observations that might also indicate that the RLS was never fully molten at any given stage. To hopefully aid the authors with their revision, I have general suggested comments, and inline comments. Overall, I think that this is a useful addition to the petrological literature, and opens some intriguing new questions as well as perhaps helping to close the textbook on perhaps more ingrained and thermodynamically poorly constrained ideas that have persisted for decades, or at least to encourage more lateral thinking.

General Comments

1. In general, I think the authors could make the paper a little bit more accessible to readers while also helping to address the 'arcane' comment above. This could start with the title, which might be impenetrable to some, and less than exciting to others. I would suggest something along the lines of "Formation of layering in the world's largest igneous intrusion through small-scale melt crystallization" or something of that nature. I leave it to the author's as to the exact terminology, but I suspect they might agree that the current title, while factual, is not necessarily going to have researchers or Bushveld protagonists scrambling to read it. On the same lines, I have specific comments about removing the criticism of being in danger of becoming arcane in the inline comments on the PDF.

We have changed the title and abstract a lot. The novelty of the idea is clearly highlighted in the new title. The inline comments in the PDF have been corrected in the new revised manuscript, and our modifications in response to each inline comment can be found in the

new attached PDF. We note also that the preprint received hundreds of reads on Researchgate within a few weeks, about half of which were full downloads, indicating that it did attract considerable attention from the community even with our initial title.

2. It would be useful if some more thermal arguments that might support their assessment are brought into the discussion. For example, the thermal aureole of the RLS is remarkably limited given that it was supposed to have been an entirely molten magma chamber at one stage. Equally, the preservation of a 'chilled margin' with komatiitic affinity is remarkable if the RLS was indeed completely molten, or even hot, as this model also argues. The authors might also wish to show more explicitly how an AFC model fits or does not fit with the lower most portion of the RLS – this is shown in the figure, but a brief clearer mention to it would help.

The entire paper is predicated on an isenthalpic model of assimilation and crystallization that explicitly conserves thermal energy, and the conditions at which this modeling was done were chosen with reference to the geotherm. The significance of the chilled margin to us is merely that it indicates that the parental magma to the entire suite was likely to have been as MgO-rich as a komatiite. The immediate local significance of the thermal gradient implied by the existence of chilled ultramafic rocks in the lower and basal zones is really too fine-scale a matter to be covered in the present paper. We can easily come up with ad hoc explanations why a komatiite might become chilled against quartzite but digressions like that will merely invite criticisms of the details of whatever scenario we put forward. We do note however that an extensive but poorly exposed contact aureole is developed at the base of the RLS, principally within the rocks of the Pretoria Group. The contact aureole extends to a distance of at least 25 km, which corresponds to an orthogonal thickness of ~ 4 km. On the other hand, the intensity and extent of thermal aureole depends on physical parameters, like thermal diffusivities or the temperature contrast, which are related to the compositions of magma and country rocks. The heat transfer between magma and wall rocks are also controlled by many factors, e.g., convection in magma, hydrothermal circulation in the country rock, and the mechanism and timescale of magma emplacement. Hence, we do not think there is a clear relationship between the aureole thickness and the melt- or mush-dominated RLS. In order to avoid some unnecessary uncertainties, we will not use the aureole thickness as supporting evidence.

3. The authors could expand on describing the chill margin with spinifex olivine and why this is a good proxy for a melt composition. If the model of a vast molten magma chamber with the same starting composition were true, then this would be reasonable. However, we are not discussing that type of a model. We are discussing a pulsed addition of melt into a complex network of crystal mush or something akin to crystal mush (i.e., in many places 99% crystal). How does one preserve a spinifex texture with 7-9 km of igneous material above that must have at least been fairly hot (maybe not hot enough to be molten, but hot). How is this preserved given the argument for the ease with which komatiite becomes contaminated? Ultimately, much of this is important, as turbulent flow is invoked at the high differential temperatures of the komatiite liquid to the crustal assimilate. This may be true at the inception of the RLS, but it cannot be true during the later phases of layering when the system has 'warmed up' and I imagine that's part of the logic requirement for polybaric crystallization as the modelling progresses.

There have been 4 whole papers to clearly describe the compositions, isotope data, texture and associated significances of chill margin with spinifex olivine (Wilson, 2012, 2015, 2017; Maier et al., 2016), and hence we just cited these works without using many words to describe it, especially given the word number limit of this journal. In the paragraphs that describe the B1-B3 magma, we have listed many evidences to support that the crustal assimilation of komatiite is a more reasonable scenario: 1) most primitive olivine and orthopyroxene observed in LZ should be crystallized from a komatiitic magma, rather than the suggested B1 magma; 2). the geochemical features (major and trace elements) of B1 magma suggested that it is derived from the crustal contamination of komatiites in Archean greenstone belts; 3). the remarkable similarities of trace-element patterns have led to suggestions that the B1 and B2-3 magmas were derived from komatiite via $> \sim 40\%$ contamination by upper and lower crust, respectively; 4) the Cr budget in the RLS require that the parental magma must have been komatiitic. In addition, the chill margins with spinifex texture were found in two different position: the BUS and the LZ. Hence, this is not a rare sample in the RLS. These evidences mentioned above are close to our assumptions for the assimilation model with a komatiite parental magma (in the same page), and hence we think there is no need to list them again.

Because the assimilation occurs in the initial stage of magma ascent and emplacement, the melt velocity and viscosity make the flow to match the turbulent flow regime, where the crystallized grains remain in suspended. But we have never said this state will be continued to the final consolidation, the assimilation degrees in our models always remain in the small values, which also guarantee the assumption of turbulence in the assimilation stage. In the manuscript, we have clearly explained that the solid will settle from the flow to form the cumulates when flow velocity drops to a low value where the flow is not turbulent. In order to make this point clearer, we added a new paragraph to describe the slumping of crystal slurries within the laminar flow regime in the revised manuscript.

Regarding the possibility that the crust will be too warm to allow vigorous convection - first, any crustal material that remains solid will be cool enough to set up very large thermal gradients against a komatiite melt; and second, we argue mainly for forced convection rather than thermal convection as the driver for turbulent flow within the komatiite.

4. The authors point out that “a hypothetical dimensionless thermodynamic black box ABC process” has been successful. This raises the point that, if the authors presented their model, perhaps as an excel spreadsheet, or maybe whatever program format they used (MELTS output), this would really be helpful to the community and would be an important contribution from the work. Can the authors do this?

The associated files (i.e., environment file and melt file) used in the AlphaMELTS, in conjunction with the excel spreadsheet, have been attached as the supplemental material of the new revised manuscript. We have even given a fairly detailed set of instructions for the commands to use in alphaMELTS to duplicate the modeling procedure, although users probably would need some familiarity with the program to reproduce everything we have done.

5. The modelling of isotopic compositions in Fig. 6 is not well expressed. Are these simply mixing models, or do they incorporate information from the MELTS modeling? The bars for the different zones are bit complicated. In general, the authors need to explain the modeling

in this figure much more clearly. In comments on the supplement PDF, I also have queries about the endmember compositions used. Using the North China Craton composition seems really odd here, given that the geographic location is very different. Are there no lower crustal xenoliths from local kimberlites of the Kaapvaal or environs that could be used instead?

More information of the modelling of isotopic compositions have been added in the revised manuscript. We describe the reasons for the selection of isotopic compositions of the mixing endmembers. The modeling itself is simply done by mass balance using the starting compositions and the masses and phase compositions we get from the MELTS output. Because the ABC model assume the equilibrium state between the melt and mineral phases, the isotopic mixing model is pertinent in this environment and adopted here. The systematic Sr-Nd isotope data for some zones are highly limited (e.g., B3), which may introduce some uncertainties. On the other hand, the recent works just focused on the laser in-situ Sr isotopic compositions of minerals in distinct zones, but the corresponding Nd isotope data have not been measured. The horizontal bars in the top right region of Fig. 6 represent the total distribution ranges of Sr isotopes collected from all measurements no matter whether they have the according Nd isotopic compositions, which may offer a more objective description for the Sr isotopic features of different zones. The existing studies on the surrounding kimberlites of Bushveld Complex and Kaapvaal Carton are focused on the mantle xenoliths, and the information about the lower crust in this region is rare. The estimated composition of the lower crust is based on sampling of 11,451 individual rock samples from the North China Craton (Gao et al., 1998), and has been widely adopted as a classical model for the global lower continental crust compositions (>860 citations, not limited to the NCC). The subsequent estimations from Rudnick & Gao (2003, 2014) also chosen this composition for middle crust by averaging mid-crustal rocks exposed in China. It is much the same approach that western scientists employed previously using crust exposed near their institutions, and is presumably as appropriate as any other estimate except for the fact that it is based on hundreds of times more data than most previous estimates. Here, we adopted the initial estimation from Gao et al. (1998) to model the assimilation occurred in the lower crust. We provided a Supplementary Figure 5 to show that the assumed composition of lower crust in this study falls into the range of the worldwide database.

6. The work could be more explicit about the broad zonal classification being oversimplified for its regional-scale utility, while the cumulate layers show numerous and complex mesoscale variations in their spatial distribution. This is a nuance that some might not catch, while simultaneously using the chromitites as markers. It is an important point, difficult to explain by a vast AFC magma chamber model, but perhaps also equally difficult to explain with an ABC model as well. For example, the presence of potholes and vast accumulations of chromitite is not explicitly modelled here.

We have added some sentences to explain the zonal classification and the presence of potholes in the revised manuscript. The issue of regional and local variations in the cumulate sequence has been expanded on in the revised manuscript. We use it in support of the idea that the chromitite-bearing macrolayers are in some ways easier to explain as sills than as layers in a predictable sequence.

However the main point of the article is not to argue solely for out-of-sequence sill emplacement to construct the RLS. As we say in the discussion, the mechanisms of delivery of crystals that we explore in this article could operate along the base of a magma chamber or

within discrete sills, but crucially do not **require** the presence of the magma chamber to occur.

We cannot model pothole formation in an article about the composition of the entire complex. Potholes are a fascinating phenomenon that we acknowledge to be somehow related to thermal erosion. To say any more than that would simply serve to distract readers from our main points and incite arguments about features that are of second or third order importance to the overall picture.

7. This brings me to the greatest area of concern surrounding the manuscript, which is the development of monomineralic layers and sulfide behavior. The formation of monomineralic layers is never expressly given in the paper, not does the modelling ever approach a 'pure magnetite' or a 'pure chromitite'. This is critical, as these layers are stratiform and span nearly the entire complex. What's more, these layers have been the subject of intense scientific scrutiny and numerous hypotheses, including an insitu origin, have been given for them. I cannot see how 'small scale melt batches', as expressed in the last paragraph of the paper, could ever achieve these accumulations, unless small scale is really quite big, and merely a relative term. These chromitites exist throughout much of the lower portion of the sequence. Magnetite and Nelsonite layers occur in the upper portions and these require explanation too. The MELTS modelling cannot achieve this, yet a major conclusion is that ABC can generate such layers, as well as the S-rich layers forming features like the Merensky Reef. I don't think this paper gets close to addressing these major features of the RLS. Presumably, the authors leave open the possibility of nearly every idea, other than formation of these layers by crystal settling from a big open and molten magma chamber. This is certainly a weak point in the paper, and will be exploited, if the authors do not 'plug the hole' here by perhaps providing some explicit examples of how the modelling relates to such features. Alternatively, they might simply wish to admit that the vast accumulation of Cr spinel and magnetite require further work in the context of the RLS.

Thanks for your insightful comments. We have mentioned the formation mechanism of the chromitite layers in RLS under the considerations of our ABC model here, but it seems to be overlooked easily. In the revised manuscript, we have clearly explained our preferred mechanism of mechanical crystal sorting from cotectic mixtures in detail and highlighted this part several times to emphasize its importance. Again, a detailed examination of the mechanisms and implications of crystal sorting would require an entire series of other papers and the best we can do is to mention it repeatedly and to cite our other recent work where the ideas were developed more thoroughly.

Most forming processes proposed to account for the chromitite layers involve variations in chemical and/or physical conditions that drive the magma to become supersaturated in chromite and remove other phases from the liquidus, resulting in the formation of monomineralic layers on the magma chamber floor via the accumulation of chromite grains. However, all of these mechanisms that make chromite as the only liquidus phase in magma chamber cannot account for the Bushveld Complex because it contains far more chromite than that can be derived from the apparent mass of the present body and also contains abundant examples of non-cotectic proportions of chromite and either pyroxene or, bizarrely, plagioclase. This problem can be solved by the crustal assimilation by komatiite in a staging reservoir (Hales & Costin, 2012) and subsequent crystal sorting of an internally equilibrated slurry, which is one of the foundations for our transcrustal assimilation models here.

Inline comments:

Because no line numbers were given, I provide inline comments on both the main manuscript and supplement PDFs – attached.

Reviewer #3 (Remarks to the Author):

It is an excellent paper that makes a convincing (and plausible argument) against the RLS (at least in its economically important (roughly) lower half) being derived by the solidification of a melt-dominated chamber. Instead, the authors argue that this part of the RLS formed from the emplacement of non-sequentially injected crystal slurries. These slurries were processed in deeper storage chambers – where they underwent differentiation by ABC (Assimilation Batch Crystallization) – prior to emplacement in the RLS. The economically important Upper Critical Zone, specifically, underwent two stages of ABC with emplacement of its feldspathic units first, followed by its pyroxenites.

Not exactly what we propose. We suggest that to form a norite you need a two-stage process wherein the first lot of ultramafic cumulates is deposited at depth along the way up to the RLS, and then the resulting supernatant liquid undergoes further cooling to deposit a second batch of plagioclase-pyroxene cumulates within the RLS. In contrast, the pyroxenitic layers can form from a single-stage ABC process.

In contrast, the upperparts of the Main Zone and Upper Zone (termed the UUMZ) were derived by ‘conventional’ AFC of large volumes of ferrobaltic magma in the chamber. The quantitative modelling that the authors use in support of their model appears to be sound and it can be reproduced using the supplementary information (though admittedly I did not have time to do this). The case the authors make is important because there is a longstanding convention (that is still perpetuated now) that the RLS is the product of crystal sorting and melt fractionation (+ recently by in situ crystallization) in a melt-dominated system. However, there is abundant (and growing) field and chemical evidence that this is not the case, and that magmas emplaced into the RLS were processed in deeper magma reservoirs and they entrained crystal cargoes during emplacement (in accord with the magma plumbing dynamics of other continental flood basalt provinces throughout geological time).

I splattered some notes in the attached pdf of the manuscript for the authors to look at. They mostly relate to the following general points and only seek to help the authors strengthen their arguments:

- The connection between the non-sequential assembly model for the Critical Zone and its economically important layers (PGE reefs + chromitites) needs to be better developed. For instance, the final sentence of the Abstract comes out of the blue and there is certainly a paragraph missing in the Discussion that should explain the following points; how are the formation of PGE reefs in the RLS explained by the melt-dominated chamber model, and how can the authors preferred model better explain their formation? It would be good to see a clear hypothesis put forward in this regard that can be thought about and tested in the future.

We have added a one-sentence recapitulation of the traditional magma mixing model for PGE reef formation, and now also cite our earlier work in which we showed that the amount of PGE in a 250-thick layer of U-type magma could generate a reef deposit if the magma

reached sulfide saturation. Since assimilation of crust by komatiite is the undisputed mechanism for the formation of magmatic sulfides in most examples, we don't feel the need to explore the details of such a process any further here. The model is explained in our cited work and is also explored in other recent papers that are too numerous to cite here. The idea that chromitite can form by mechanical sorting of cotectic crystal mixtures is mentioned several times now in the revised text and is the subject of at least three other articles we cite that explore the topic more deeply. Citations to these hypotheses are provided and cannot be explained in any greater detail in this short article.

- I made a few comments in places that refer to the research that Kruger presented regarding a plethora of papers in the 1980s/early 90s using initial $87\text{Sr}/86\text{Sr}$ ratios. Kruger (1994 and 2005) summarised these data to argue that the RLS formed from two stages (Integration and Differentiation Stages). I do not think the authors have done this work justice in the relevant parts of the manuscript – and, if one was being highly critical, the authors model could just be considered as a more detailed explanation of Kruger's work. I think the important point the authors need to make is that Kruger perhaps considered his model in the context of a melt-dominated chamber - with little detail about komatiitic melts processing crust and generating isotopically modified slurries – maybe the authors did make this point in the article, but I did not pick it up?

The pioneering works of Kruger (1994 and 2005) undoubtedly underpinned the subsequent researches on the RLS. In this manuscript, the basic situation introduction of RLS is also based on the two-stage model (Integration and Differentiation) from Kruger, and cited his classical paper published in 1994. Because of the style of citation in this journal, it looks like that we have not mentioned Kruger's work, but actually, the associated reference [23] occurs in many positions. In the new revised manuscript, we have also added his paper (published at 2005) into the reference lists, and highlights Kruger's indispensable contributions to the researches on the RLS and layered intrusions.

However the key difference between his ideas and ours is that he imagines the need for repeated reversals in isotopic composition of a large body of resident magma (which we did say) to explain the variations in composition of the cumulates, whereas we argue that these differences result from different pathways to the site of deposition.

- I do not think there was any mention of either inter- and intra-crystal isotopic disequilibrium that has been documented in the RLS (see data in Prevec et al. 2005; Chutas et al. 2012; Roelofse & Ashwal, 2012; and lots of other recent papers using laser Sr isotope data in plagioclase). It would be good to see these data considered at some point and how it could be explained in the context of a transcrustal magmatic system. They might also want to consider mentioning the role that intercumulus liquid percolation played in the isotopic signatures of the cumulates (+ minerals) as they are in favour of the existence of thick piles of mush during RLS solidification.

Many thanks for this important proposal. We have acknowledged the inter- and intracrystalline Sr isotopic heterogeneity throughout the RLS, and exhibited the total ranges of Sr isotope data (including the laser Sr isotope data) in the horizontal bars at the upper right region in Figure 6a. We also recognize that the Sr isotopic heterogeneity and the zoned plagioclase grains may have resulted from the multiple, isotopically distinct influxes of melt percolating through the cumulate framework during RLS solidification. Recent work on the

Rum layered intrusion (Hepworth et al., 2020, NG) also proposed that the highly localized reactive melt percolation in crystal mush not only drive the Sr isotopic heterogeneity, but also trigger the formation of precious-metal-bearing chromitite layers. This model is mentioned in the revised manuscript. Furthermore, we have added some commentary on the fact that batches of crystals amassed along a dynamic pathway through highly variable crust might produce major element compositions and mineral modes that are fairly constant while retaining highly variable isotopic compositions due to mixing of contaminants of different isotopic composition. We also responded to this issue in our response to the other reviewers.

- This may be beyond the scope of the present article, but it is perhaps something to consider: The discrepancy between zircon Hf isotopes (constant throughout the RLS; after Zirakparvar et al. 2014) and the (?) common presence of isotopic disequilibrium in the major cumulate-forming silicate phases (many solution and laser papers on this) is puzzling. Is it possible that the homogeneous Hf isotope signature of zircon is a product of a isotopically homogeneous carrier melt that entrained ABC processed slurries from depth? Or could this homogeneous Hf signature be explained by RLS-level contamination (recent Zeh paper in JPet)? We've been thinking about this problem here in Joburg and we would be happy to see the authors present a solution to this problem in the context of their model.

Good suggestions. The homogeneity of the zircon $\epsilon\text{Hf}_{(2.06 \text{ Ga})}$ values for RLS is a confusing phenomenon. First of all, the Hf isotopic compositions of the contaminants for the parental magma of RLS are poorly understood. Zirakparvar et al. (2014) used the zircon Hf isotopic ratios of Limpopo Belt rocks (all from the Zeh's previous works) as the contaminants, but such a composition is not representative of surrounding rocks of the RLS. On the other hand, no thermodynamic considerations were included into their simple mixing model for the major elements in the assimilated magma. Hence, we are suspicious of Zirakparvar's explanation that the zircon Hf isotopic features of RLS mostly come from the SCLM. More recently, the Zeh paper in JPet (2020) obtained many zircon Hf isotope data from the surrounding rocks of RLS, and found that the RLS show essentially the same ranges of $\epsilon\text{Hf}_{(2.055 \text{ Ga})}$ as those of zircon grains from the surrounding quartzite, metapelitic rocks, granite, granophyres and volcanics. Hence, the assimilation processes in the RLS may not drive a highly variable Hf isotopic compositions, which are also suggested to be homogenized during the post-cumulate stage. The Sr and Hf isotopic compositions can be decoupled by their different partitioning behaviors, distinct concentrations and isotopic compositions of crustal materials, and variable sensitivities to fluid infiltration in post-cumulate stage, which has also been explained clearly in Zeh's paper. Here, we tend to give more support to the interpretation from Zeh et al. (2020), and have mentioned it in the revised manuscript.

- According to mineral compositional profiles for the entire RLS (e.g., in Cawthorn, 2015) – it appears that pyroxenites of the UCZ have lower Mg# (in opx or ol) compared to parts of the LCZ or LZ. Assuming (here comes arguably another misplaced convention of the RLS...) that there is a smooth upward decrease in the Mg# of ferromagnesian minerals in sub-UUMZ parts of the RLS, then how do the authors reconcile this with their non-sequential emplacement model? Would the mineral compositional profiles not look a bit more random in this part of the RLS? It would be useful to see this addressed in the discussion.

In short: Just as people have wanted to see cyclicity in the Peridotite zone of the Stillwater Complex when a dispassionate statistical analysis shows comprehensively that cyclicity is completely absent, the pattern of decreasing Mg# (what Rais Latypov calls "order from chaos") has often been described as orderly when it is not orderly at all. There is no smooth

upward evolution in the lower part of the RLS. It is a mess. The only way to account for the wild oscillations in every geochemical parameter imaginable in the lower 2/3 of the RLS is to appeal to a rather stochastic process of intrusion of different magmas, whether one adheres to the classic magma chamber model or not. If the intrusions of new magma occur within a large magma chamber, then truly enormous amounts of material need to be added each time the composition of the cumulates is seen to change, to swing the composition of the whole system back and forth, whereas if each layer is treated as a single intrusive event, there is no need to reconstitute the isotopic, trace element, and major element composition of the entire intrusion each time there is a change in the crystallizing assemblage seen in the cumulate pile. The contrast with the UUMZ, which does mostly follow predicted patterns from FC, could not be starker.

The generalized trends of vertical decreases of Mg# in orthopyroxene grains seem to be valid for the whole RLS, but here Cawthorn (2015) just used the Mg# ranges of orthopyroxene in a certain range of height to exhibit the trends (the **(a)** in following diagram), which strongly neglects a lot of details. When the vertical section zoomed into the region of LZ and UCZ, Cawthorn (2015) also observed a much more complicated pattern of the Mg# of orthopyroxene in detail for the Eastern Lobe (**(b)**) and Western Lobe (**(c)**). Here, only minor net differentiation trends can be observed in the ultramafic rocks of the LZ and UCZ. Many fluctuations and even the large-scale reversals (increasing Mg# with the upward height) are common in the LZ-UCZ sections. On the other hand, the Mg# of orthopyroxene in the UUMZ quickly decreases from 65 to 5 due to the differentiation within ~2000 m thickness, while in contrast, the Mg# just decreases from 90 to 65 for the total thickness (~5000-7000 m) of the LZ, UCZ, LCZ and LMZ, which suggests a distinct process relative to the differentiation in UUMZ.

[REDACTED]

In our transcrustal assimilation models here, the modelled assimilation degrees increase from Dun-LZ (~17%) to Px-LCZ (~34%), corresponding to the reductions of relative proportion of the ultramafic component due to the addition of upper crusts, which in turn, drives the decrease of Mg# values of olivine and orthopyroxene. On the other hand, the sill

emplacement and associated lateral propagation are mostly attributed to the deflection of vertical dykes into horizontal sills, especially when the crystal-laden magma achieved its neutral buoyancy. Because the densities of olivine and orthopyroxene decrease with the drops of their Mg# number, the emplacement of Px-LCZ loaded with less-Mg# minerals has a great possibility to occur above the dense LZ, which can account the general decrease trends of Mg# in the sub-UUMZ parts of RLS. However, the deflection of ascending magma is also influenced by the material toughness and elastic mismatch, which refers to whether the surrounding cumulate remains in solidification, semi-consolidated state or primary formation of crystal framework with high porosity. Hence, the intercross between different mushy macrolayers are very common, which may account for the field observations and the highly variable Mg# of orthopyroxene in the vertical section.

We have added this part into the discussion section of the revised manuscript.

REVIEWERS' COMMENTS

Reviewer #1 (Remarks to the Author):

The authors have addressed the comments I made on the previous manuscript version to my satisfaction. Thanks for this, and for the interesting discussion points that they included in the rebuttal letter. Overall I think this is a timely, novel and provocative study that will generate lots of interest in the petrology and wider Earth Sciences community. I recommend the paper now be accepted.

Brian O'Driscoll

Reviewer #3 (Remarks to the Author):

I have read the rebuttal letter and the revised manuscript and I am happy that the authors have addressed all of the matters raised in the previous round of reviews. They have adapted their manuscript accordingly and have included more detailed aspects of their model (particularly with regards to monomineralic layer formation and the origin of inter- and intra-crystal isotope disequilibrium).

They also now give a stronger argument as to why their preferred model (mushy chamber) better explains the origin of the stratiform PGE reefs and chromitite layers - and have clearly shown why the old fashioned model (melt-dominated chamber) has limitations in this respect. There is a whole new paragraph about this.

The title is also better. It is more reflective of the significant findings in this paper.

All of the AFC and ABC results now seem to be appended, including a step-by-step guide. This is very useful.

I have little more to add.